# Systematic transcriptional analysis of human cell lines for gene expression landscape and tumor representation

Han Jin [1,5], Cheng Zhang [1,5], Martin Zwahlen [1], Kalle von Feilitzen [1], Max Karlsson [1], Mengnan Shi [1], Meng Yuan[1], Xiya Song[1], Xiangyu Li [1], Hong Yang [1], Hasan Turkez[2], Linn Fagerberg [1], Mathias Uhlén [1,3] ✉ & Adil Mardinoglu [1,4] ✉

Cell lines are valuable resources as model for human biology and translational medicine. It is thus important to explore the concordance between the expression in various cell lines vis-à-vis human native and disease tissues. In this study, we investigate the expression of all human protein-coding genes in more than 1,000 human cell lines representing 27 cancer types by a genome-wide transcriptomics analysis. The cell line gene expression is compared with the corresponding profiles in various tissues, organs, single-cell types and cancers. Here, we present the expression for each cell line and give guidance for the most appropriate cell line for a given experimental study. In addition, we explore the cancer-related pathway and cytokine activity of the cell lines to aid human biology studies and drug development projects. All data are presented in an open access cell line section of the Human Protein Atlas to facilitate the exploration of all human protein-coding genes across these cell lines.

With the sharp increase in the application of (multi-)omics technology and computational modeling of biological systems, novel therapeutic targets and drug candidates for human diseases have been identified through in silico analysis[1-4]. Cell line experiments, which often serve as the first step in translational research for clinics, are extensively used in verifying potential drug candidates and newly identified targets. In the past decades, more than ten thousand cell lines were established for various studies with the same goal: to simulate the actual disease mechanism. Due to metabolic alterations in cell line models[5] and reduced biological availability in humans, drug candidates may still have a high chance of failing in the in vivo studies and clinical trials, even though they are successfully tested in vitro based on more than one cell line. Therefore, whether and to what extent each cell line model can reflect a disease phenotype,

especially after establishing immortalized and stabilized cell lines from primary cells and culturing them in vitro, remains a critical question in life science.

Tremendous efforts have been made to reveal the molecular characteristics and genetic specificity of cell lines, including the transcriptomics data (RNA-sequencing; RNA-seq) of human cell lines in the Cancer Cell Line Encyclopedia (CCLE)[6,7] and in the cell line section of the Human Protein Atlas (HPA) project[8]. The transcriptomics data generated by these studies enabled the comparison of global gene expression between the cell lines and the corresponding tumor based on the transcriptomics data (RNA-seq) in The Cancer Genome Atlas (TCGA) project. Despite a few studies that have attempted to map cancer cell lines to the corresponding diseases based on the molecular signatures[9-11], which facilitates the selection

[1]Science for Life Laboratory, Department of Protein Science, KTH Royal Institute of Technology, Stockholm, Sweden. [2]Department of Medical Biology, Faculty of Medicine, Atatürk University, Erzurum, Turkey. [3]Department of Neuroscience, Karolinska Institute, Stockholm, Sweden. [4]Centre for Host-Microbiome Interactions, Faculty of Dentistry, Oral & Craniofacial Sciences, King's College London, London, UK. [5]These authors contributed equally: Han Jin, Cheng Zhang. ✉e-mail: mathias.uhlen@scilifelab.se; adilm@scilifelab.se

of appropriate cell lines for cancer research, these studies did not further extend their comparison of cell line gene expression to normal tissues and individual cell types—a critical step which will help us to understand the characteristics and representative of cell lines to their origin[12].

In this study, we first analyze the transcriptomics data of more than 1000 human cell lines and compare the global gene expression landscape of the human cell lines with the gene expression data from human tissues, tumors, and single-cell types. Second, we systematically evaluate if and which cell lines are representative of specific cancer types. Next, we analyze the cell line characteristics at pathway and cytokine levels for providing a deeper understanding of the cell phenotype as well as generating a reference state for a better design of in vitro experiments. We finally provide all the information in the cell line section of the HPA portal (https://v22.proteinatlas.org/humanproteome/cell+line) in an open access format.

## Results

### Establishment of the HPA cell line section

Previously, the HPA cell line section included transcriptomics data of 45 cancer cell lines from 20 cancer types, 16 non-cancerous cell lines, and 8 uncategorized cell lines in which the disease types have not yet been classified (marked as "Unknown"; see Supplementary Data 1)[8,13]. The CCLE 2019 cell line database[6] provides the transcriptomics data of a total of 1,019 cell lines collected from 35 sites in the human body (Fig. 1a) including 973 primary and metastatic cancer cell lines derived from 26 cancer types, 44 non-cancerous cell lines (mostly fibroblasts; see Fig. 1b), and 2 cell lines with uncategorized disease types (denoted as "unknown") (Supplementary Fig. 1 and Supplementary Data 2). We found that transcriptomics data of 33 cancer cell lines were present both in the HPA and CCLE databases (Fig. 1a). To create a comprehensive transcriptomic atlas of human cell lines, we added the transcriptomic data for 1,019 cell lines from the CCLE 2019[6] to the HPA cell line section, which extended the number of non-cancerous cell lines

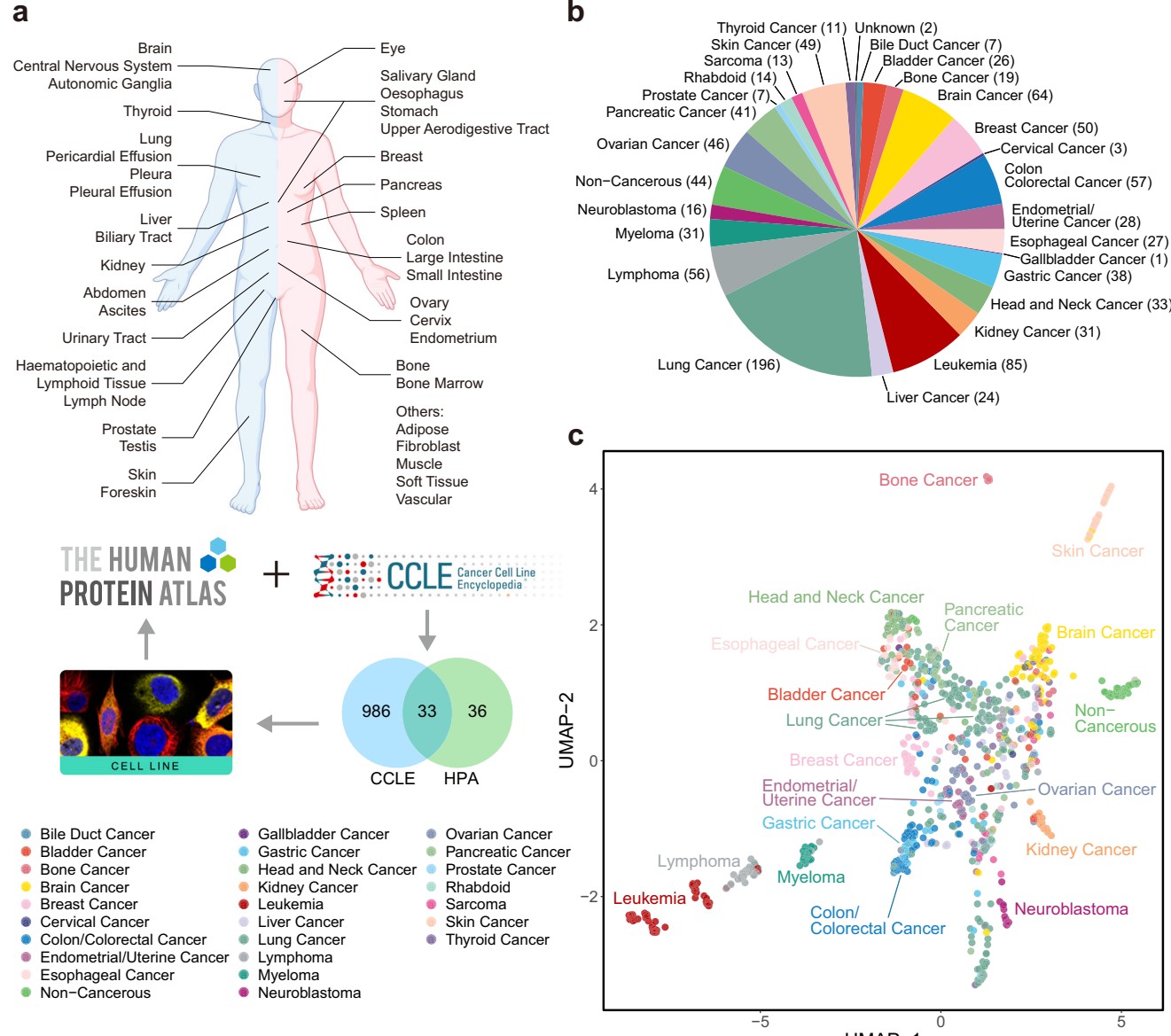

**Fig. 1 | Establishment of the open access cell line section. a** RNA-seq data of 1,019 cell lines from the CCLE[6] and 69 cell lines from the HPA (33 overlapped) were jointly analyzed in this study. These cell lines were collected from around 40 different human body sites. Figure created with BioRender.com. **b** Pie chart showing the number of cell lines per disease in the CCLE 2019 database. **c** UMAP plot showing the relationship between the 1,019 CCLE cell lines color-coded by primary disease. Source data are provided as a Source Data file.

from 16 to 60, cancer cell lines from 45 to 985, uncategorized cell lines from 8 to 10, and cancer types from 20 to 27 (Fig. 1a).

We retrieved the raw data from the CCLE and HPA database and calculated the expression values nTPM (TMM-normalized pTPM; where pTPM is the TPM for protein-coding genes, see Methods; TMM, trimmed mean of M values; TPM, transcripts per million) of these cell lines using the same pipeline for RNA-sequencing data preprocessing. We then applied principal component analysis to extract 398 PCs preserving 80% of the total variance in the full dataset, followed by Uniform Manifold Approximation and Projection (UMAP)[14] to visualize sample distribution. We found that all cell lines were evenly mixed and almost every pair of the common cell lines both in the CCLE and HPA databases stayed tightly close with each other (Supplementary Fig. 2). To further confirm the closeness of these common cell lines, based on the 398 PCs, we calculated the Euclidean distance from CCLE to HPA cell lines, and found that for the 33 common cell lines, 32 have their counterpart as the first neighbor in HPA. We further examined the relationship between all the analyzed cell lines (CCLE + HPA) by Spearman's correlation and found that 33 pairs of the common cell lines between CCLE and HPA were clustered together, except for a single cell line, U-251MG (Supplementary Fig. 3a). The high correlation of the transcriptomics data of the common cell lines suggested that the batch effects can hardly be observed between the two different datasets. This may be attributed to the fact that cell lines generally exhibit more stable characteristics than animal models where heterogeneity must be taken into account. Considering the high consistency of transcriptomics data in the CCLE and HPA datasets, we incorporated the CCLE transcriptomics data into the HPA database without further processing for batch correction.

As shown in Fig. 1c, cell lines from blood cancers such as lymphoma, leukemia, and myeloma (all derived from hematopoietic and lymphoid cell systems), as well as bone cancer and skin cancer, formed separated clusters from the major cluster for all the other cancer types, suggesting distinct characteristics of these cell lines to the other cancer cell lines. Within the major cancer cell line cluster, several cancer types such as neuroblastoma, kidney cancer, and breast cancer showed clear cancer-specific sub-clusters, implicating that these cancer cell lines may also have their own features distinguishing them from the others. In addition, cell lines of gastric cancer and colon/colorectal cancer which both belong to the upper digestive system also displayed close relationships with each other, highlighting the similarity between these two types of cancer cell lines. Indeed, this was also demonstrated by the significantly higher correlations between the cell lines from the same disease than the ones from different diseases (Supplementary Fig. 3b). In addition, using the top 5000 most variable genes, 74% and 68% of the cell lines can be correctly classified (5-fold cross-validation) for their cancer types by logistic regression and random forests, respectively. Taken together, these results suggested that cell lines preserve the cancer phenotype at the transcriptional level.

**Classification of protein-coding genes in human cancer cell lines**
Next, we explored the gene expression distribution across all the CCLE (*n* = 1019) or HPA (*n* = 69) cell lines and independently categorized the genes based on their specificity of expression in the HPA and CCLE datasets (see Table 1 for criteria). We found that the number of non-detectable genes (638) in the CCLE dataset is strikingly lower than the number of non-detectable genes (2,066) in the HPA dataset (Fig. 2a). The number of genes expressed specifically in a single cell line was also lower in the CCLE dataset compared to the HPA dataset (283 vs. 856). Hence, by incorporating the CCLE dataset into the HPA database to significantly increase the total number of cell lines, we found that the number of cell line-specific or not detected genes became smaller, suggesting that the integration of the two cell line datasets may reflect a broader coverage of gene expression. In addition, we found that

5,366 genes (with 5,209 overlapping with those 6,799 genes in HPA; Fig. 2a) are expressed in all 1,019 CCLE cell lines (nTPM > 1), suggesting these genes are essential in the cell line models.

Notably, we found that the number of these "housekeeping" genes is indeed significantly higher than the number of essential genes identified based on the genome-wide CRISPR-Cas9 screening[15,16]. In this context, we interrogated the 5,366 genes expressed in all 1,019 CCLE cell lines with the previously reported 1,912 essential genes based on CRISPR gene screening results (DepMap 22Q2, CRISPR_common_n_essentials.csv from https://depmap.org/portal/download/all/)[15], and found that 1,614 genes were overlapped between two different datasets. Based on gene set overrepresentation analysis (GSOA)[17], these genes were strongly associated with DNA replication and nuclear division, suggesting these genes are indispensable for cell cycle progression and cell proliferation (Fig. 2c, Supplementary Data 3). Of note, we found that 3,751 genes are specifically present in the CCLE dataset (Fig. 2b), and their functions were significantly associated with the basic biological processes in different cellular compartments (Fig. 2c and Supplementary Data 3). Hence, we observed that silencing of these genes may not directly induce cell death; the wide expression of these genes, however, indicates that these genes are fundamental for maintaining basic cellular functions.

Focusing on cancer research, here we integrated 45 HPA cancer cell lines with 973 CCLE cancer cell lines (33 pairs of common cell lines) to obtain a total of 985 cancer cell lines to study cancer-specific cell line characteristics. Specifically, twelve cancer cell lines and one cancer type (i.e., testis cancer) were only available in the HPA dataset but not in the CCLE dataset. By averaging the gene expression nTPM in cell lines within each cancer type (defined by the primary disease of a cancer cell line), 985 cancer cell lines were aggregated into 27 groups based on cancer types. Hereafter, we termed CLD (cell line disease) to represent cell lines grouped by the same cancer type. Subsequently, protein-coding genes were categorized into five different groups according to the expression specificity in CLDs (Table 1), including (i) CLD-enriched genes with at least fourfold higher expression levels (based on nTPM values) in one CLD as compared with any other analyzed CLD; (ii) group-enriched genes with enriched expression in a few CLDs (2 to 10); (iii) CLD-enhanced genes with only moderately elevated expression; (iv) low CLD specificity genes showing elevated expression in at least one of the analyzed CLDs; and (v) not detected genes at the CLD level. Similarly, we analyzed the TCGA transcriptomics dataset including 6,082 primary tumors from 26 cohorts with high tumor purity scores[18] (>0.7), and applied gene expression specificity classification to the TCGA dataset at the cohort level, which enabled the comparison of the gene expression specificity between CLDs and the TCGA cohorts.

Based on the criteria in Table 1, a total of 1,340 enriched genes showed at least fourfold higher expression in one CLD as compared to any other CLD (Fig. 2d). In addition, 927 genes were defined as group-enriched genes since these genes have elevated expression in up to 10 CLDs. Another 3,511 genes were categorized as enhanced genes in CLDs since they showed at least a fourfold increase in expression compared to the average. Taking these into account, a total of 5,778 genes were elevated in at least one of the 27 CLDs. Meanwhile, 11,770 and 2,542 genes showed low CLD specificity and low expression across 27 CLDs, respectively. The number of genes differently distributed in CLDs is similar to what was observed in the 26 TCGA cohorts (Fig. 2d), with 13,331 (66.47%) genes being classified as exactly the same category between CLDs and TCGA cohorts, suggesting a high similarity of gene expression distribution and specificity between cell line and TCGA datasets.

At the cancer type level, CLDs were hierarchically clustered based on the averaged expression of cell lines from the same cancer type, resulting in three distinct clusters of CLDs (Fig. 2e). In line with early analysis in Fig. 1c, three blood cancers (i.e., myeloma, lymphoma, and

**Table 1 | Categories used for cell line gene expression specificity and distribution**

| | Category | Description |
|---|---|---|
| Specificity | Enriched | A single CLD (cell line gene expression averaged by disease) has 4 times higher expression than any other CLD |
| | Group enriched | 2-10 CLDs have 4 times higher expression than any other CLD |
| | Enhanced | One or more CLDs have 4 times higher expression than the average of all other CLDs |
| | Low specificity | The gene does not belong to any of the above categories and is detected above cut-off (nTPM = 1) in at least one CLD |
| | Not detected | All CLDs have an expression value less than 1 |
| Distribution | Detected in single | Detected in a single cell line above cut-off (nTPM = 1) |
| | Detected in some | Detected in more than one but less than one third of the cell lines |
| | Detected in many | Detected in at least a third but not all cell lines |
| | Detected in all | Detected above cut-off (nTPM = 1) in all cell lines |
| | Not detected | All cell lines have an expression value less than 1 |

For gene expression specificity, gene expression in cell lines was averaged by cancer types.

leukemia) formed an independent cluster, with a high number of elevated genes compared to the others. The second major cluster consisted of CLDs mainly derived from major human internal organs such as the lung, colon, and breast. Notably, gastric and colon cancers again showed a very close relationship in the hierarchical dendrogram. The last cluster of CLDs involved 10 cancer types including liver and kidney cancers.

### Comparison between cancer cell lines and tissues/tumors/cell types

To evaluate if cell lines are representative of the corresponding tumors, we compared the CLD-enriched genes with the corresponding TCGA cohort-enriched genes, meanwhile including tissue-enriched and single-cell type-enriched genes defined by the HPA[19,20] for comparison. The tissue-enriched and single-cell type-enriched genes were derived from 56 tissues and 51 cell types from 13 human tissues (both are non-disease), respectively, with similar gene expression specificity strategies applied. As shown in Fig. 3a, a high concordance can be observed between cell line cancers and their corresponding tissues, TCGA cohorts, and single-cell types. For example, genes enriched in liver cancer cell lines were strongly overrepresented in liver tissue, TCGA-LIHC (liver cancer), and hepatocytes (Fig. 3a). After examination, we found that more than half of the enriched genes in liver cancer cell lines were also enriched in liver tissue, TCGA-LIHC cohort, and hepatocytes (Fig. 3b), suggesting a high concordance of enriched genes between liver cancer cell lines and the corresponding tissue, tumor, and cell type. Another example is leukemia, where the genes that were enriched in the cell line cancers were found strongly overrepresented in TCGA leukemia cohort (LAML), bone marrow, lymphoid tissue, erythroid cells, granulocytes, and T-cells (Fig. 3a). In addition, genes enriched in neuroblastoma, testis, skin, prostate, and breast cancers can also be mapped to the corresponding tissues, cancers, or single-cell types (Fig. 3a), suggesting a high concordance between CLD-enriched genes and the enriched genes obtained from bulk tissues, tumors, and single-cell types.

Apart from the overrepresentation analysis of the enriched genes, additionally, we deployed a correlation-based analysis to evaluate the consistency between the genome-wide transcriptomic expression of cell line cancer types and the one from the corresponding TCGA cancer cohorts. In brief, gene expression of the TCGA samples from the same cohort was averaged, and the correlations between CLDs and TCGA cohorts were calculated by Spearman's ρ. As shown in Fig. 3c, most of the TCGA cohorts can be matched to the corresponding CLDs, such as LAML to leukemia, SKCM to skin cancer, READ and COAD to colon/colorectal cancer, and HNSC to head and neck cancer. Taken together, despite that cell lines from the same primary disease are to some extent heterogeneous (primary vs. metastatic, different collecting sites, etc.), both overrepresentation analysis of enriched genes

and correlation analysis based on all protein-coding genes suggested a high concordance between CLDs and the corresponding TCGA cohorts.

We then compared primary and metastatic cell lines to investigate the changes in gene expression induced by metastasis. In terms of primary cell lines, the high concordance between CLDs and tissues/cancers/single-cell types is maintained, with similar significant levels of overrepresentation (Supplementary Fig. 4, lower). Although the significance level slightly decreased, when only including metastatic cell lines for analysis, the levels of overrepresentation between the cell line cancers and the corresponding tissues/cancers/single-cell types were still high (adj. P-value < 1E-30 for the most significant ones in Supplementary Fig. 4 upper part), implicating that metastatic cell lines preserve the majority of the transcriptomic features of their primary cancer types. We also observed that the metastatic cell lines (from the same disease or same site) showed lower sample-wise correlations than the primary cell lines (Fig. 3d), which suggested a higher heterogeneity among metastatic cell lines in general. In addition, metastatic cell lines from the same site showed lower sample-wise correlations compared to metastatic cell lines from the same primary cancer type, which implicated that the origin of the cancer cell lines has a greater effect on their overall transcriptomic expression as compared to the environment in the metastatic site.

### Prioritizing cell lines as models for human cancer

Given the high concordance of gene expression between cell lines and the corresponding tumors (Fig. 3a, c), we evaluated how similar a cell line is to its corresponding cancer type and prioritized the most representative cell lines for the 26 TCGA cancer cohorts (Supplementary Data 4). In order to do that, we evaluated the cell line similarity to its bulk tumor by two different approaches: (1) correlation-based, i.e., Spearman's correlation between the gene expression of cell lines and the averaged expression of the bulk tumor samples from the same cancer type; and (2) enrichment-based, i.e., the enrichment of the expression level of the TCGA cohort elevated genes (i.e., the union of the enriched, group enriched and enhanced genes of a TCGA cohort, also named as "cohort signature"; see Supplementary Fig. 5) in a cell line. For the second approach, in brief, we calculated the genome-wide relative expression of every cell line by comparing it to the baseline expression (i.e., the average expression of all CLDs), and evaluated the expression of the TCGA cohort elevated genes in the disease-matched cell lines by gene set enrichment analysis (GSEA) (see Methods for details). It is expected that a good cell line model should have a high expression of the corresponding TCGA cohort elevated genes (indicated by a high normalized enrichment score, NES).

In fact, the correlation-based and enrichment-based approaches agree well with each other, with the cell line highly ranked by one measurement also ranked high by the other (Fig. 4). We then

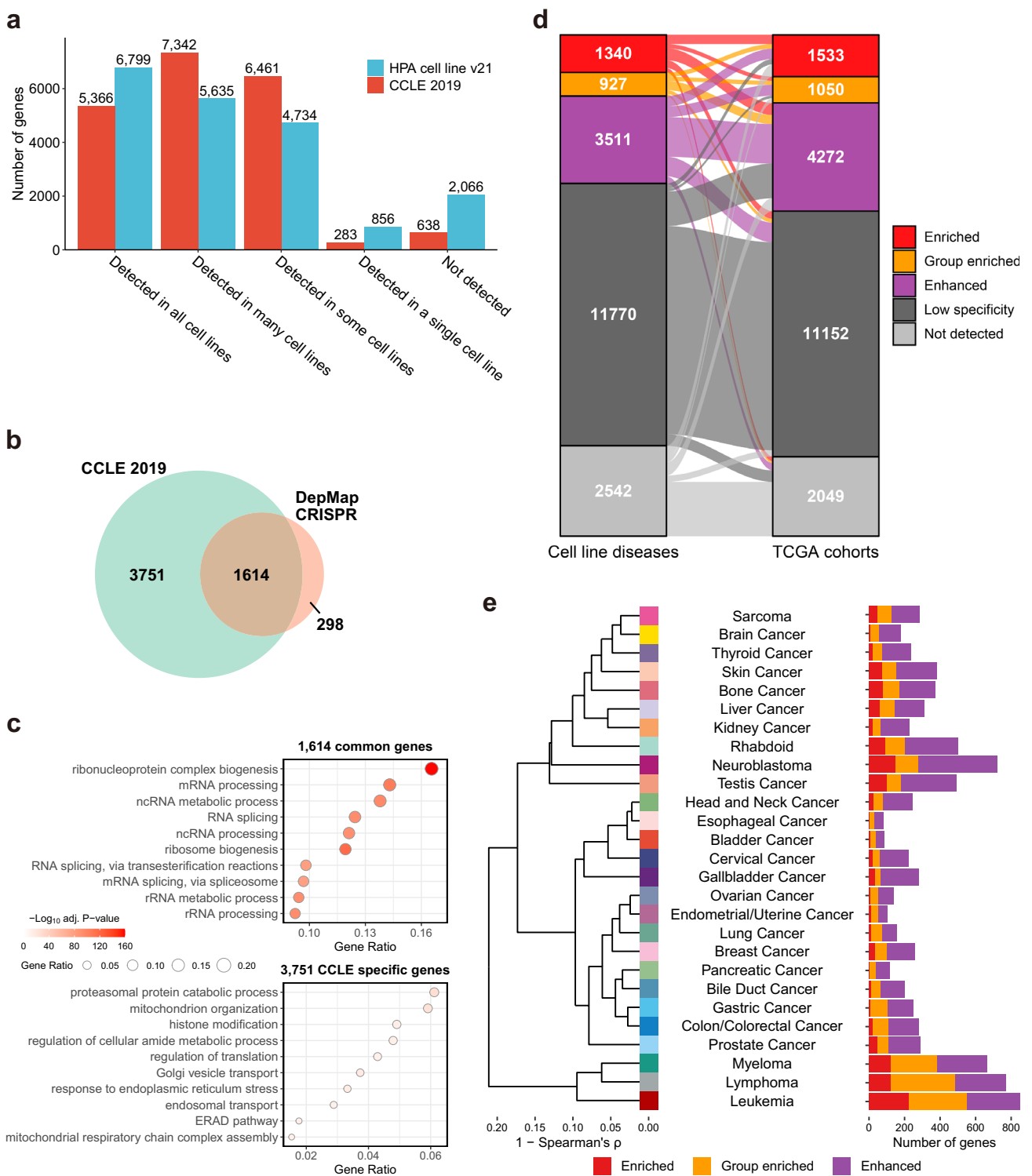

**Fig. 2 | Genome-wide classification of all protein-coding genes in human cancer cell lines. a** Comparison of the distribution of gene expression in the CCLE 2019 and HPA cell lines (version 21). **b** Venn diagram showing the overlap between 5,366 genes (5,365 with gene symbols) detected in all 1,019 cell lines in CCLE 2019 and the 1,912 essential genes based on CRISPR gene screening results. **c** GSOA (hypergeometric testing) of the 1,614 overlapped genes and the 3,751 CCLE-specific genes in (**b**). *P*-values were adjusted based on the Benjamini-Hochberg procedure. Ten highly significant GO terms are selectively shown. **d** Alluvial diagrams showing the number of genes of respective specificity categories for CLDs vs. TCGA cohorts. **e** (Left) Hierarchical clustering of the 27 CLDs based on the correlation of averaged expression profiles of cell lines for the same cancer type. (Right) Bar plot showing the number of elevated genes in each cell line cancer type. Source data are provided as a Source Data file.

integrated the two approaches above and prioritized the top 5 (including tied for fifth) highest-ranked cell lines as candidate models for each TCGA cohort (Fig. 4 and Supplementary Data 5), resulting in a total of 114 cell lines selected as good candidates for cancer research.

To validate the selected candidates, we compared the results with a previous study using 5,000 most variable genes for correlation-based cell line selection (TCGA-110-CL; *n* = 100 selected cell lines for the same cohorts analyzed in this study)[9]. We found that 65 of our selections

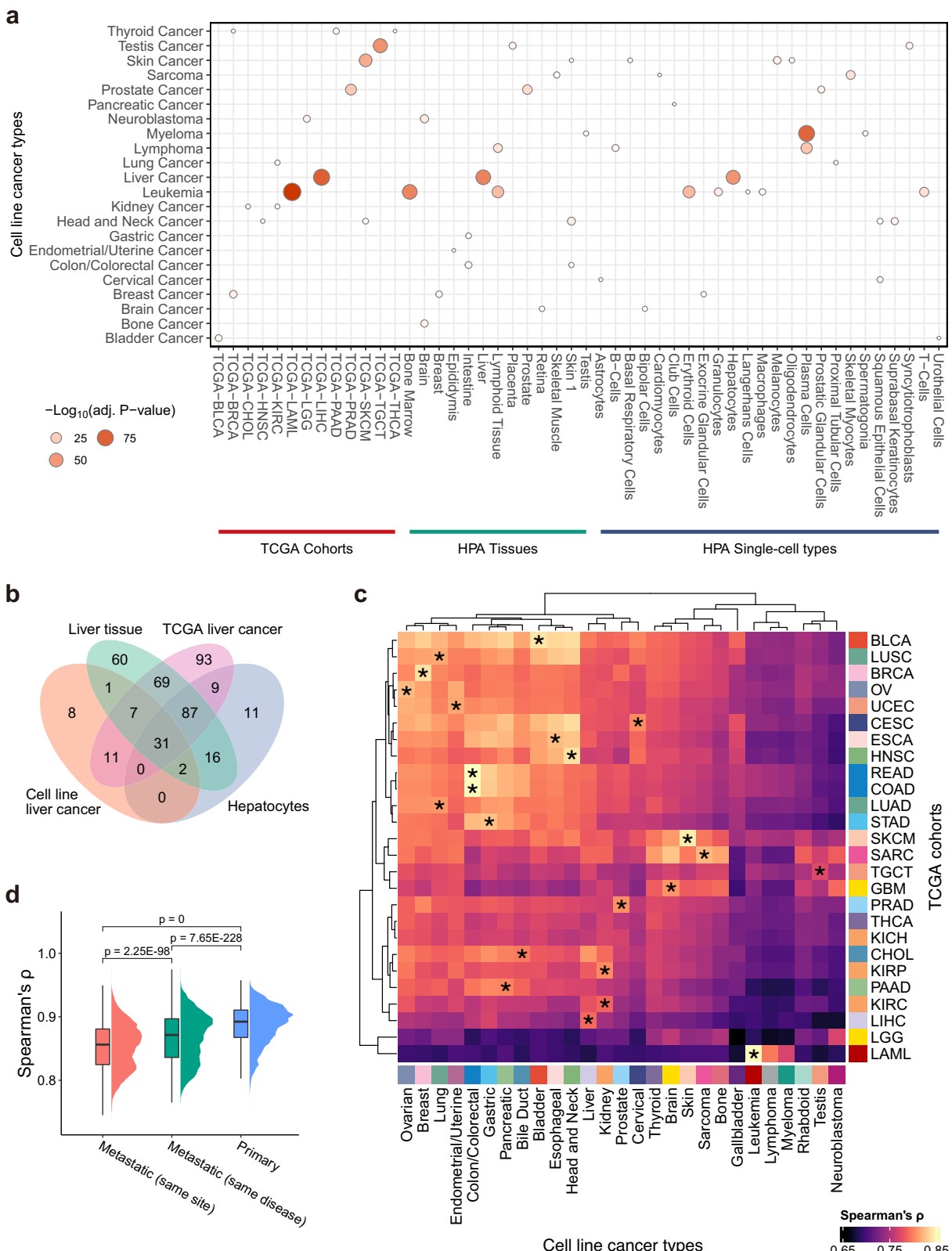

were also included in the TCGA-110-CL panel, with a p-value of 1.55E-44 based on hypergeometric testing (Supplementary Fig. 6a), under-pinning the validity of the selected cell lines by our approach. Of note, our study relied on all protein-coding genes rather than a subset of genes for correlation analysis, which could provide a more compre-hensive and precise evaluation of the similarity between cell lines and tumors.

For some cancer types, e.g., liver cancer (LIHC), the commonly used cell lines such as Hep-G2 and Huh-7 for in vitro study were selected as the best candidates out of the total 24 liver cancer cell lines. We then specifically inspected the expression of the TCGA-LIHC sig-nature (867 genes elevated in TCGA-LIHC cohorts compared to the others) in the 24 liver cancer cell lines, and found that the expression of the TCGA-LIHC signature was generally higher in the top-ranked cell

**Fig. 3 | Comparison of transcriptomics between cancer cell lines and TCGA cohorts, HPA tissues, and single-cell types. a** Dot plot showing the significance (estimated by hypergeometric testing) of the overlapping genes between the enriched genes in CLDs (y-axis) and TCGA cohorts, HPA tissues, and single-cell types (x-axis). P-values were adjusted based on the Benjamini-Hochberg procedure. Non-significant overlaps (adj. P-value > 0.05) are not shown in the figure, and CLDs that are not significantly overlapped with any TCGA cohorts, tissues, single-cell types, or the other way around, are removed. **b** Venn diagram showing the intersected genes between the enriched genes in cell line-based liver cancer and the TCGA liver cancer, HPA-analyzed liver tissue, and hepatocytes in single-cell type analysis. **c** Correlation between the CLDs and TCGA cohorts calculated based on the average expression per CLD and TCGA cohort. For each CLD, we used one-sided one-sample Wilcoxon signed-rank test to investigate if the correlations to its unmatched TCGA cohorts were significantly lower than the correlation to its matched TCGA cohort. Based on the information in Supplementary Data 4,

26 statistical tests were performed. *$P < 0.05$. **d** Correlation between cell lines by different categorizations. Primary: correlations between primary cell lines. Correlations between cell lines were calculated per cancer type and were summarized ($n = 7,189$ correlations). Metastatic−same disease: correlations between metastatic cell lines. Correlations between cell lines were calculated per cancer type and were summarized ($n = 6,864$ correlations). Metastatic−same site: correlations between metastatic cell lines. Correlations between cell lines were calculated per sample collection site and were summarized ($n = 7,627$ correlations). Statistical significance was evaluated by two-sided Wilcoxon rank-sum test. The lower, middle, and upper hinges correspond to the 25th, 50th, and 75th percentiles. The upper whisker extends from the hinge to the largest value no further than 1.5 * IQR from the hinge (where IQR is the inter-quartile range, or distance between the first and third quartiles). The lower whisker extends from the hinge to the smallest value at most 1.5 * IQR of the hinge. Source data are provided as a Source Data file.

lines than in the lowly ranked cell lines (Fig. 5a). For instance, the Albumin (*ALB*) gene—one of the TCGA-LIHC signature genes, which is also a hallmark gene of liver functions[21], was highly expressed in the top-ranked liver cancer cell lines (Fig. 5b), distinguishing prioritized cell lines from lowly ranked cell lines. Indeed, based on GSOA, the LIHC signature was associated with basic metabolic functions in liver (Fig. 5c), suggesting that the expression of the LIHC signature is mechanistically critical for liver cancer cell lines. Moreover, we plotted the correlation of gene expression between three representative cell lines (two highly ranked cell lines, Huh-7 and Hep-G2; and one lowly ranked cell line, SNU-398) and the TCGA-LIHC cohort (Fig. 5d). We observed that the two highly ranked cell lines demonstrated a high correlation to the TCGA-LIHC ($\rho = 0.774$ and $0.772$, respectively) while the lowly ranked cell line, SNU-398, had a relatively low correlation ($\rho = 0.665$). Meanwhile, the TCGA-LIHC signature was also found relatively lowly-expressed in the SNU-398 cell line, suggesting that the hallmark of TCGA-LIHC might be weaker in this cell line. Indeed, a previous study reported that some cell lines presented an undifferentiated state and may be derived from undifferentiated tumor[10]. In our ranking list for the 24 liver cancer cell lines, only the SNU-886 (ranked 12th) was reported as "undifferentiated" in the first 12 cell lines, but 9 of the last 12 cell lines were reported as "undifferentiated". The imbalance between highly- and lowly-ranked cell lines regarding their differentiated state possibly explains why some cell lines are lowly prioritized. Taken together, these results demonstrated the validity of the use of the approaches for cell line prioritization.

In addition, for each TCGA cohort, we also applied our cell line prioritization algorithm to evaluate cell line fidelity at the pathologic stage and molecular subtype levels (Supplementary Data 6 and 7). We then validated our results by comparing them with the TCGA-110-CL cell lines ranked for subtypes based on two TCGA cohorts (the same subtype categorization as in this study), and found that the correlations between the rank of the cell lines in this study and in the TCGA-110-CL were generally high (all *p*-value < 0.05; Supplementary Fig. 6b). When focusing on the stage and subtype of the TCGA-LIHC cohort, we found that Huh-7 was ranked as the first 5 cell lines and outperformed Hep-G2 in all conditions except for tumor stage II. For this particular tumor stage, in the total 24 liver cancer cell lines, the Huh-7 was ranked 6th and the Hep-G2 was ranked 2nd. While showing a relatively low performance than Hep-G2, the Huh-7 still outperformed most of the cell lines, suggesting a broader scope of usage, thus the Huh-7 could be a good candidate for the research for all the tumor stages and molecular subtypes.

**Cancer-related pathway and cytokine activity in human cell lines**
To evaluate the cell lines from a functional perspective, we inferred the cancer-related pathway and cytokine activity for all 1,055 unique cell lines included in this study. PROGENy[22] and CytoSig[23] were employed for this analysis, and they allowed us to investigate the association

between the cell line expression profiles and 14 cancer-related pathways as well as 43 cytokine signaling cascades (Supplementary Data 8), respectively. Results of the pathway and cytokine activity were presented as z-scores, representing the strength of the corresponding signal relative to the average across all the 1,055 unique cell lines (Fig. 5a and Supplementary Data 8; see Methods for details). We found that most of the non-cancerous cell lines, i.e., fibroblasts, displayed high levels of fibrotic pathway TGF-β and cytokines TGF-β1/-β3 activity (Fig. 6a)[24]. Meanwhile, the MAPK signaling pathway—the key pathway and therapeutic target in melanoma[25], was highly activated in skin cancer (Fig. 6a). In addition, hematopoietic and lymphoid cells, such as myeloma, lymphoma, and leukemia cell lines, showed high levels of apoptosis-related signaling including the tumor necrosis factor-related apoptosis-inducing ligand (TRAIL) and TWEAK cytokine (Fig. 6a), suggestive of a particular role of the TRAIL signaling and TWEAK cytokine in blood cancer.

We also projected the results of pathway and cytokine analysis for all cell lines on a UMAP plot to investigate whether cell lines with the same cancer of origin showed similar pathway and cytokine features. In both analyses, blood cancer cell lines (i.e., myeloma, lymphoma, and leukemia) formed a distant cluster that can be easily separated from the other cell lines (Supplementary Fig. 7a, b), which is in line with the expression-based UMAP visualization (Fig. 1c). The pathway and cytokine activity was validated by a previously published independent dataset[26] (Genentech; $n = 610$, with 461 overlapping with the cell lines analyzed in this study). Based on the mean squared error (MSE) of the pathway and cytokine activity between each pair of the cell lines from this study and the Genentech datasets, the 461 common cell lines showed significantly lower MSE than all the pairs of the non-common cell lines at both pathway and cytokine levels (Supplementary Fig. 7c, d). Further, we also calculated the pathway and cytokine activity for the TCGA cohorts, and similarly, prioritized cell lines based on the MSE to the matched TCGA cohorts (Supplementary Data 9). We found a significant overlap ($n = 31$; hypergeometric testing *p*-value = 4.14E-07) of the prioritized cell lines between pathway/cytokine analysis and transcriptome-wide analysis (Fig. 4). Taken together, these results demonstrated that these two computational tools were able to extract meaningful pathway and cytokine signaling levels from the cell line transcriptomics datasets.

The pathway and cytokine profiles of the 1,055 cell lines also provided the opportunity to investigate the relationship between pathways and cytokines. Based on the pathway analysis of 1,055 cell lines, pro-inflammatory pathways TNF-α and NF-κB[27] were highly correlated with each other (Supplementary Fig. 7e). Meanwhile, the hypoxia pathway which promotes angiogenesis and metabolic reprogramming was strongly associated with TGF-β, a pathway playing important roles in tissue repair (Supplementary Fig. 7e). At the cytokine level, pro-inflammatory cytokines such as IFN-γ, TNF-α, tumor necrosis factor-like weak inducer of apoptosis (TWEAK), and

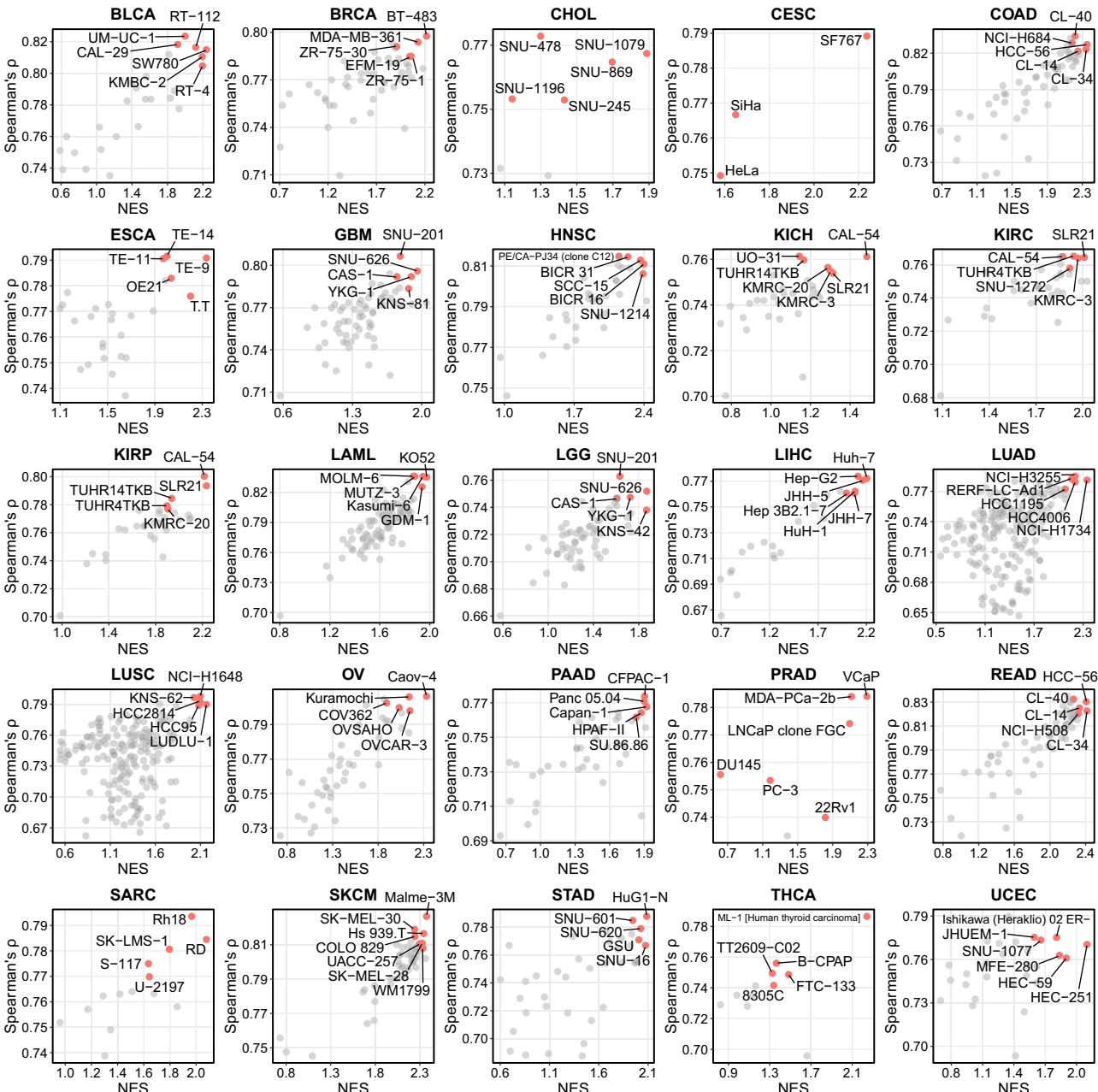

**Fig. 4 | Prioritizing cancer cell lines as models for human cancer.** For each TCGA cohort, the top five highest-ranked cell lines for the same cancer type were prioritized based on the integration of correlation-based rank (*y*-axis) and GSEA-based rank (*x*-axis; NES normalized enrichment score). Names of the selected cell lines are shown (top-5 including tied for fifth for each cohort), with these cell lines in the dot plot highlighted in red. The correlation between the testis cancer cell line SuSa and the TCGA testis cancer cohort TGCT is 0.77, with an NES of 1.97 (adj. *P*-value = 9.37E-81) for the TCGA testis cohort elevated genes in the SuSa cell line (data not shown in this figure, see Supplementary Data 5). Source data are provided as a Source Data file.

IL-1α/−1β were highly correlated with each other (Supplementary Fig. 7e). Pathways and cytokines also showed nice correlations correspondingly. For example, the TGF-β pathway was highly correlated with the TGF-β1 and TGF-β3 cytokines, whereas pro-inflammatory pathways TNF-α and NFκB were highly correlated with pro-inflammatory cytokines TNF-α and IL-1α (Fig. 6b). In addition, the immune-related pathway JAK-STAT was highly correlated with type 1 interferon (IFN1), IFN-γ, and IL-27 (Fig. 6b), the latter of which can induce IFN-γ and regulate the immune system via the JAK-STAT signaling[28].

We can also investigate the pathway and cytokine signaling levels from a cell line-centric perspective. As an example, here we investigated the pathway and cytokine activity in three individual cell lines of breast cancer. By comparing the highest prioritized cell line BT-483 with the commonly used cell line MCF-7 for breast cancer, we found both cell lines showed a high estrogen signaling activity (Fig. 7a), which is in line with the report that both cell lines are estrogen receptor (ER)-positive[29]. The ER-negative breast cancer cell line BT-549 did not show a significantly high level of estrogen signaling (Fig. 7a) but demonstrated a high pro-inflammatory (TNF-α, IL-1α/−1β) and a low anti-inflammatory response (IL-4) relative to BT-483 and MCF-7 (Fig. 7b). The levels of pathway activity were similar between BT-483 and MCF-7, with the MAPK pathway significantly inhibited in only BT-483 (Fig. 7a). But the two cell lines demonstrated different cytokine levels (Fig. 7b). Interestingly, at the cytokine level, our results identified elevated BMP4 signaling in these three and other breast cancer cell lines

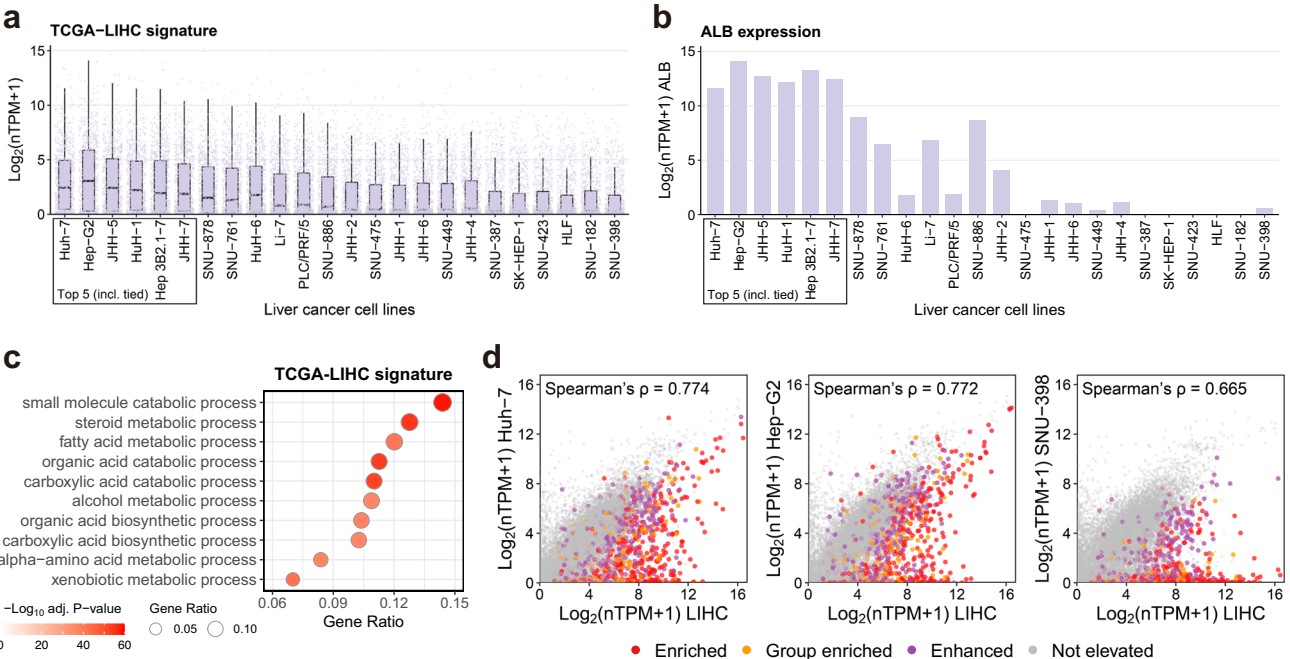

**Fig. 5 | Selected cell lines for liver cancer. a** The expression of the TCGA-LIHC elevated genes (signature; $n = 867$) in liver cancer cell lines. Each dot in a cell line is a TCGA-LIHC elevated gene, and the distribution is presented by a boxplot. The lower, middle, and upper hinges correspond to the 25th, 50th, and 75th percentiles. The upper whisker extends from the hinge to the largest value no further than 1.5 * IQR from the hinge (where IQR is the inter-quartile range, or distance between the first and third quartiles). The lower whisker extends from the hinge to the smallest value at most 1.5 * IQR of the hinge. **b** The expression of the *ALB* gene in liver cancer cell lines. **c** GSOA (hypergeometric testing) of the TCGA-LIHC signature. *P*-values were adjusted based on the Benjamini-Hochberg procedure. Ten highly significant GO terms are selectively shown. **d** Dot plots showing the correlation between the expression of TCGA-LIHC and three liver cancer cell lines Huh-7, Hep-G2, and SNU-398, with the TCGA-LIHC elevated genes color-coded. Source data are provided as a Source Data file.

(Figs. 6a and 7b), which was previously reported as a therapeutic target in the ER-positive breast cancer[30].

To further underpin the depiction of the cell line characteristics by pathway and cytokine analyses, we compared the 1st rank cell line VCaP with PC-3, both of which are common prostate cancer cell lines used in drug repositioning[31]. As an important feature, prostatic adenocarcinoma cells express androgen receptor (AR)[32], and this can be observed in our analysis as demonstrated by the high level of androgen signaling in VCaP (Fig. 7c). However, the PC-3 did not show this feature and displayed different characteristics to VCaP at both pathway and cytokine levels (Fig. 7c, d). This suggested that this cell line may not be suitable for testing the drugs for androgen suppression therapy for prostate cancer[33,34]. Taken together, the pathway and cytokine analyses provided a complementary and interpretable view of the cell line characteristics, and it will be a great resource for the design of in vitro experiments for drug-target validation.

## Discussion

In the last few years, several studies have been launched to align cancer cell lines to disease tumors based on (multiple) omics profiling[9–11,35–38]. While these studies have substantially deepened our understanding of the cell line representative of human cancers, they mostly lack answering a fundamental question, that is, how well is the representative of cell lines to the corresponding tissues, tumors, and cell types. In this study, by investigating the expression of 20,090 protein-coding genes in around 1,000 human cancer cell lines, we comprehensively compared gene expression of cancer cell line models with the corresponding tissues, tumors, and single-cell types. In general, cell lines exhibited high consistency with the matched cancer types, with the disease signature maintained in the corresponding cell lines. Given the high concordance between cell lines and the corresponding tissues, tumors, and single-cell types, based on two evaluation metrics,

we prioritized individual cell lines for different cancer types based on the TCGA cohort definition. Compared to the previous methods which often rely on batch correction and manifold alignment between transcriptomics of cell line and TCGA data, our approach adopts two metrics to measure (1) the global similarity between cell line and TCGA cohort (correlation-based); and (2) the preservation of disease signature in cancer cell lines (GSEA-based). Especially the adoption of the GSEA, which showed discriminative in estimating fundamental tumor biological processes in cell lines (Fig. 5c), has not yet been investigated in the previous correlation-based study[9]. Indeed, the results of the two metrics concur with each other, demonstrating the validity of our approach for cell line scoring. The prioritized cell lines displayed remarkable differences from the commonly used cell lines in the NCI-60 panel and the LINCS L1000 drug repositioning platform[31], suggesting a need of improvement of the cell line panel for in vitro study.

As a showcase, we explored the details of the selection procedure in liver cancer cell lines and elucidated the reasons why the commonly used cell lines Huh-7 and Hep-G2 outperform the others for liver cancer studies. Results showed non-ignorable differences between 24 liver cancer cell lines, in terms of the correlation (similarity) to the corresponding TCGA cohort LIHC, and the preservation of the elevated genes in the TCGA-LIHC cohort. This suggested that cell lines, even derived from the same primary diseases, are rather heterogeneous with diverse phenotypes. The major differences between cell lines could be induced by the donors, the processes during the establishment of the cell line models, and the culture procedure, however highlighting the importance of choosing the best cell line models for cell experiments.

In some cases, the selection of the most appropriate cell line may not completely follow the computational prioritization given in this study. For example, for drug-target selection, researchers may want to know if a specific gene, cytokine, or pathway is highly expressed in a

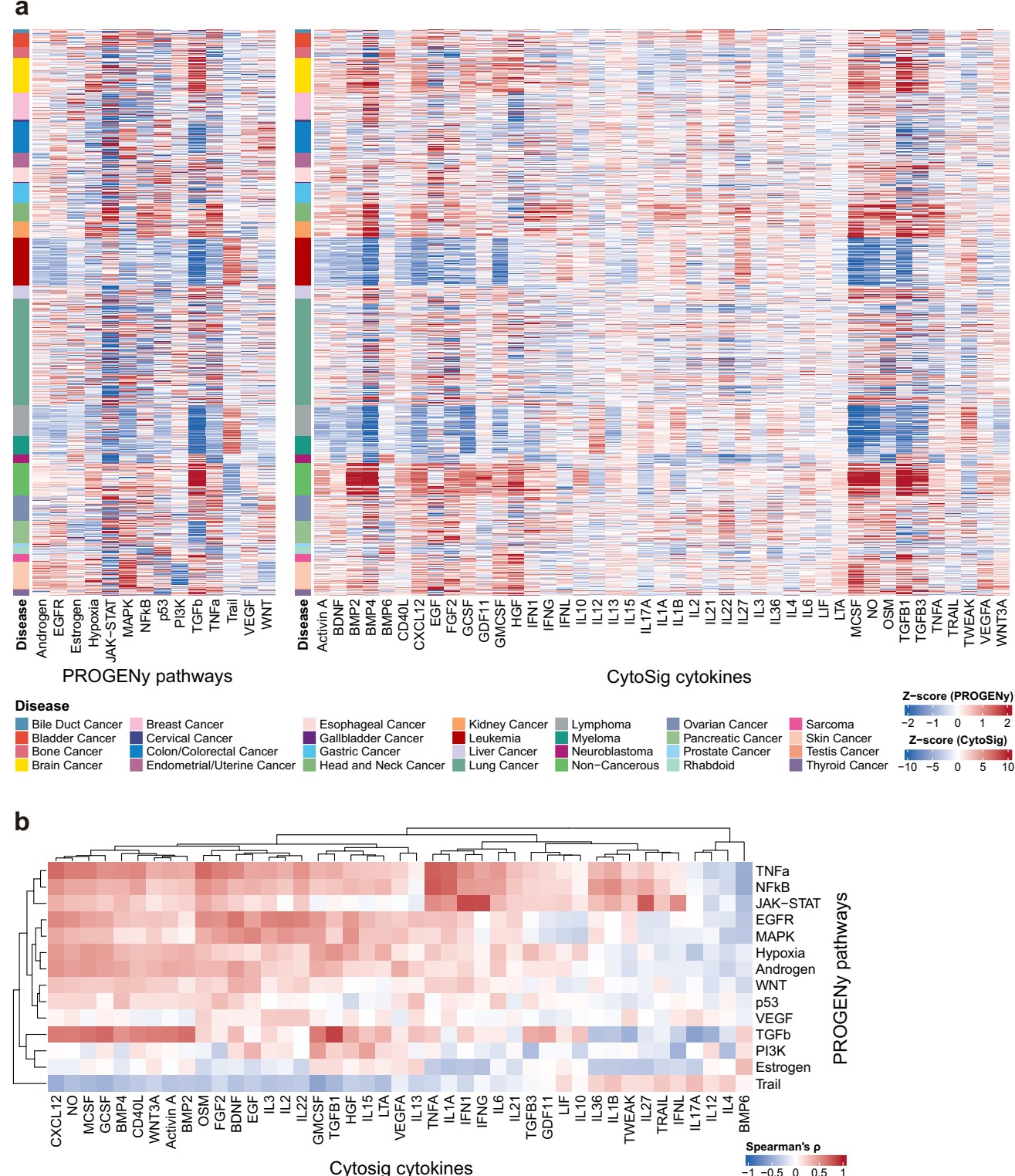

**Fig. 6 | Cancer-related pathway and cytokine activity in human cell lines.**
**a** Activity scores (presented as *z*-score) of the 14 PROGENy pathways (left) and the 43 CytoSig cytokines (right) for the 1,055 unique cell lines in this study.

**b** Correlations between the 14 PROGENy pathways and the 43 CytoSig cytokines based on the 1,055 analyzed cell lines. Source data are provided as a Source Data file.

cell line. To help interpret cell line characteristics, we calculated the pathway and cytokine activity scores. As demonstrated in several prostate and breast cancer cell lines for their AR-/ER-specificity and different levels of pro-/anti-inflammatory cytokines, these results can preciously depict the phenotypic features of the cell lines. As a complement, recent studies have enabled large-scale proteomic profiling in human cancer cell lines[39,40], providing direct evidence of the expression and translation of the biomarkers for the disease phenotype. Integrating cell line proteomics and metabolomics[41] will undoubtedly benefit the comprehensive depiction of cell line characteristics. Nevertheless, since our approaches for pathway and cytokine inference were based on biodata mining from publicly available

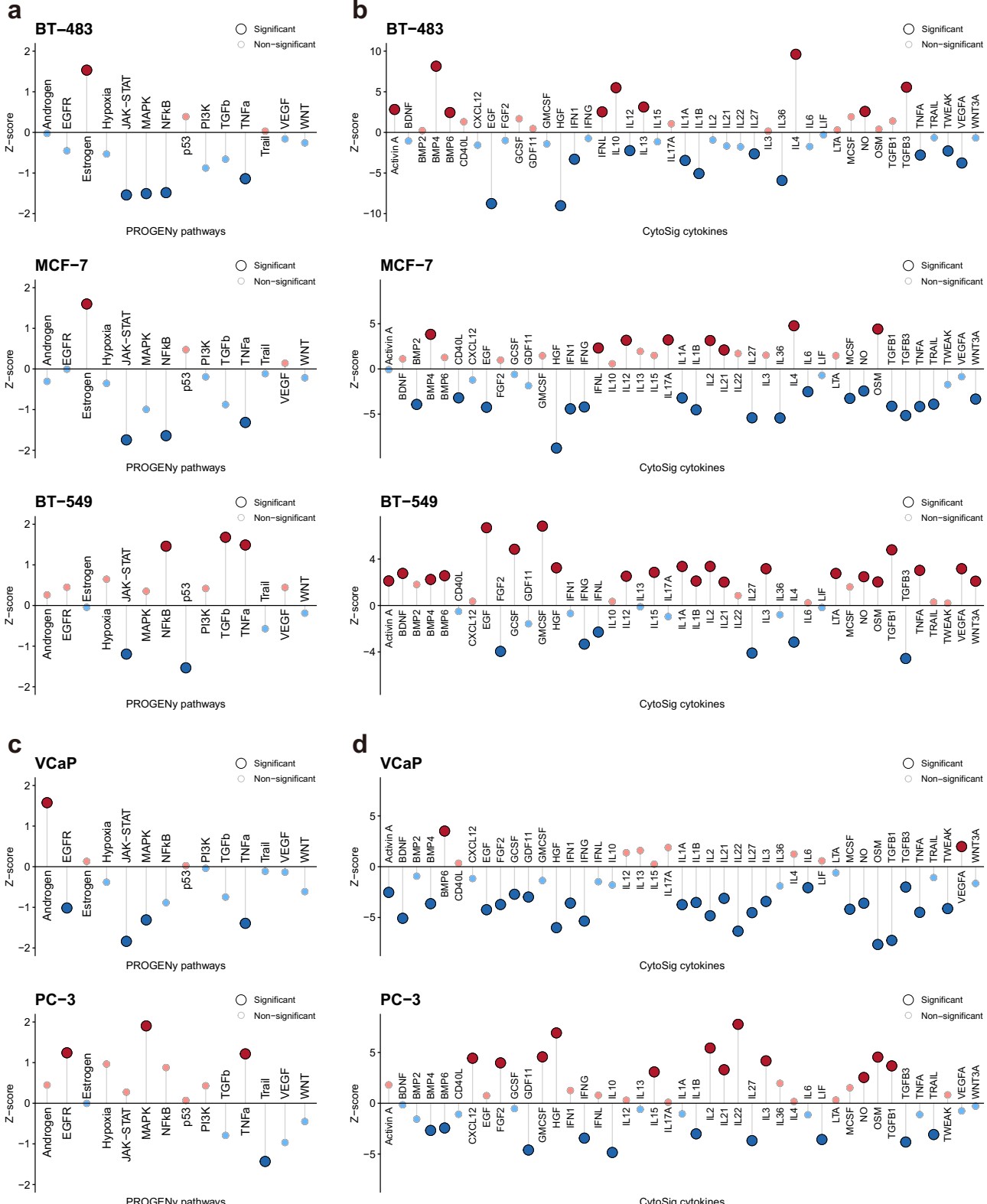

**Fig. 7 | Cancer-related pathway and cytokine activity in human breast cancer and prostate cancer cell lines. a–d** The levels of the PROGENy pathways and Cytosig cytokines for the human breast cancer cell line BT-483, MCF-7, and BT-549 are presented in (**a**) and (**b**), respectively, and for the human prostate cancer cell line VCaP and PC-3 are presented in (**c**) and (**d**), respectively. Source data are provided as a Source Data file.

transcriptomics, these results are important resources for cell line comparison that could not be derived from other omics platforms.

For all the cell lines analyzed in this study, we processed the gene expression normalization using the same pipeline applied to the other HPA resources, and present the normalized expression nTPM of these cell lines in the HPA database. Based on our previous analysis[13], the TMM normalized gene expression can minimize the batch effects caused by sampling and different technology platforms, especially

when a small number of genes are very highly expressed, facilitating the sample-wise comparison between gene expression in cell lines and tissues, organs, single-cell types, and cancer biopsies. In addition, key features of the cell lines have been demonstrated by the pathway and cytokine analysis and will be presented aside to provide a deep understanding of cell line phenotype and to aid the better design of preclinical experiments. Furthermore, we provided RRID for the cell lines in the HPA database, linking them to the information page of the Cellosaurus database[42]. This will help users to bypass the selection of mislabeled or transdifferentiated cell lines due to historical reasons[35]. By incorporating more than 1,000 transcriptional profiles of human cell lines in the HPA database, the newly launched cell line section facilitates the exploration and comparison of the expression of human protein-coding genes across cell lines and HPA tissues[19], tumors[43], single-cell types[20], and blood immune cells[13].

We here report that cell lines derived from metastatic cancers from the same tissue of origin are more similar to each other as compared to the cell lines metastasized to the same site from different origins. This suggests that although thought to be largely dedifferentiated, tumor cells keep some key features of their original cell types. This reinforces the idea that the treatment strategy for metastatic cancer should be designed based on the primary site and emphasizes the importance of the identification of cancers of the unknown primary site. Future works would focus on multi-omics analysis of cell lines using genomics and recently released proteomics[39], metabolomics[41] and drug response datasets[44–46] to comprehensively evaluate and expand the main findings in this study.

Despite advanced cell culture techniques such as 3D culture which enable a better simulation of the true in vivo environment[47], cell line models still face inherent limitations. For example, in most cases, a maximum of two cell lines can be co-cultured together, limiting the study of cellular interactions between different cell types as a whole system, the latter of which plays an important role in the tumor microenvironment and progression. Furthermore, as extensively evaluated by the recent single-cell sequencing studies, tumor cell type composition is rather heterogeneous, which also complicates the design of the in vitro preclinical experiments. Advanced cancer models such as patient-derived xenografts and organoids[48] may overcome this pitfall. Nevertheless, our study demonstrated that cell lines, as a single cell type, preserve the most discriminative and informative signatures of the corresponding cancer types, thus could be properly used as a starting point for cancer research with an appropriate selection of cell lines, the latter of which could much rely on the newly launched cell line section of the Human Protein Atlas—a comprehensive resource for the characterization and exploration of human cell lines.

## Methods

### Cell line annotation and categorization

A total of 1,019 cell lines from CCLE[6] were included in this study. CCLE cell lines were annotated based on the publicly available DepMap 2022Q2 annotation file (https://depmap.org/portal/download/all/) (Supplementary Data 2). All the cell lines were uniquely annotated by the Cellosaurus Research Resource Identifier (RRID) (https://www.cellosaurus.org/)[42]. Based on the DepMap annotation, cell lines were categorized by primary disease (i.e., cancer type), primary or metastasis, lineage, and sample collection site. This assigned the 1,019 cell lines into 26 cancer types and one non-cancerous group consists of mostly fibroblast cell lines. Two cell lines without disease information were denoted as "Unknown".

A total of 69 cell lines from HPA were included in this study. HPA cell lines were annotated by unique internal ID, RRID, cell line name, tissue, and origin. Additional publicly available information (including sex) obtained from the DepMap 2022Q2 annotation file was added to extend the HPA cell line annotation (Supplementary Data 1).

### RNA-seq data preparation and processing

The SRA files of the 1,019 cell lines from the CCLE 2019 data[6] were downloaded from the GEO using SRA Toolkit (v2.11.3) and were subsequently converted into raw fastq files. The RNA-seq data processing followed the same pipeline as the HPA project. Transcript expression levels were quantified by mapping sequences to the human reference genome GRCh38.p13 cDNA using Kallisto (v0.46.2)[49]. Based on the Ensembl version 103 annotation, the transcript abundances were aggregated into gene level as transcripts per million (TPM) by tximport (v1.22.0)[50] without the inclusion of non-protein-coding transcripts. This resulted in a total of 20,090 protein-coding genes included for further analysis, and their expression was converted into pTPM (i.e., TPM for protein-coding genes) by scaling the sum of the TPM to 1 million per sample. The pTPM expression was normalized by trimmed means of M (TMM)[51] using the tmm function provided in the R package NOISeq (v2.38.0)[52] with a median column as the reference, with the parameters doWeighting = T and logratioTrim = 0.3, and the resulting expression was denoted as nTPM.

The 69 human cell lines from the HPA database[13] were processed using the same pipeline.

To compare cell line transcriptional characteristics with TCGA cancers, a total of 9,476 RNA-seq transcriptomics profiles (count and TPM values) of primary tumors and primary blood-derived cancers from 27 TCGA cohorts were retrieved using the R package TCGAbiolinks (v2.27.2)[53]. Tumor purity of the TCGA tumors was estimated based on the ESTIMATE[18] algorithm. TCGA samples with a tumor purity score <0.7 were considered highly infiltrated by immune and stromal cells and were thus removed from the analysis, resulting in 6,082 qualified samples from 26 TCGA cohorts for downstream analysis (Supplementary Data 4). Based on the pipeline described above for cell line RNA-seq data, gene expression nTPM values were calculated from TPM for each TCGA sample.

### Machine learning prediction

Based on the nTPM expression of HPA + CCLE cell lines, logistic regression (solver = "lbfgs") and random forests (number of estimators = 100) were used to test the ability of the expression data in correctly classifying cell line disease types. The test was performed under stratified 5-fold cross-validation with 100 repeats, hence, only a disease type that has no less than five cell lines was included in this analysis ($n$ = 1076 cell lines for 26 disease labels including "non-cancerous"). The top 5000 most variable genes were selected and scaled before classification. The analysis was performed by the scikit-learn package (v1.0.2; https://scikit-learn.org/stable/index.html) in Python (v3.9.7).

### Gene expression landscape

Based on the normalized expression nTPM, genes were categorized into five groups for their expression distribution, i.e., how many cell lines a specific gene is expressed in (see Table 1 for criteria). This was done independently on the 1,019 CCLE and 69 HPA cell lines. To understand if a gene is highly expressed in one specific CLD, gene expression of cell lines was summarized per CLD (denoted by the primary disease) by average expression, followed by gene categorization for disease specificity (see Table 1 for criteria). This was done on the 973 CCLE cell lines (excluding 44 non-cancerous and 2 cell lines without disease annotation) plus 45 HPA cancer cell lines. Of these cancer cell lines, 33 pairs were overlapped between HPA and CCLE, and their nTPM expression was combined by average value. Based on the aggregated gene expression of 27 CLDs from the 985 cancer cell lines, genes were categorized into five groups for their expression specificity in CLDs, i.e., if a gene shows higher expression in one or a group of CLDs (see Table 1 for criteria).

In addition, CLDs were hierarchically clustered based on the distance converted from Spearman's correlation coefficient between cancer types (1−Spearman's ρ). The agglomeration method was set as

"complete-linkage" for hierarchical clustering. The same hierarchical clustering was applied on all analyzed 1,019 CCLE and 69 HPA cell lines.

The same approach for gene specificity categorization was applied to the gene expression data of the HPA tissue[19] and single-cell type[20] based on the HPA version 21, as well as TCGA cohorts analyzed in this manuscript. Details can be found accordingly in our previously published papers as well as in the HPA website resources (https://www.proteinatlas.org/about/download).

The CLD-enriched genes were compared with the tissue-enriched, TCGA cohort-enriched, and single-cell type-enriched genes by hypergeometric testing for significance, with the adjusted P-values corrected by the Benjamini-Hochberg procedure. In addition, functional analysis of gene sets was performed by gene set overrepresentation analysis (GSOA) based on the Gene Ontology biological process terms by the R package clusterProfiler (v4.2.2)[17].

### Cell line prioritization for cancer type

The 985 cancer cell lines from both HPA and CCLE were analyzed for their representability of the corresponding TCGA cohorts (Supplementary Data 4). The similarity between cell lines and the corresponding TCGA cohorts was estimated by two different approaches.

We calculated gene-level Spearman's correlation coefficient ($\rho$) between the cancer cell lines and their corresponding TCGA cohorts for the first. For this, for each gene in a TCGA cohort, the nTPM values were averaged per cohort. Then, for each TCGA cohort, Spearman's $\rho$ was calculated based on the TCGA averaged nTPM values and the nTPM values of the disease-matched cell lines based on the common 20,056 protein-coding genes.

For the second approach, we calculated the enrichment of the TCGA cohort elevated gene (i.e., the union of enriched, group-enriched, and enhanced genes in the TCGA cohort) in cell lines by gene set enrichment analysis (GSEA). The concept is that genes that have an elevated expression in a TCGA cohort can be considered as the cohort signature and their high expression should be reflected by cell line models. To test this, similar to the approach where we calculated gene specificity, for the 27 CLDs, gene expression was averaged per disease, resulting in the mean expression for each of the 27 CLDs. Then, the average expression per disease was further averaged as the disease baseline expression. After that, for every cell line, we calculated the fold change of every gene relative to the disease baseline expression, followed by the $\log_2$ transformation of the fold change. Finally, for each cell line, gene $\log_2$ fold changes were sorted from high to low, followed by the GSEA of the TCGA cohort elevated genes against the sorted gene list. It is expected that cell lines showing high concordance to the matched TCGA cohort should present high $\log_2$ fold changes of the TCGA cohort elevated genes relative to the disease baseline expression. The results were represented as the normalized enrichment score (NES), with a positive value showing high consistency between a cell line and a disease-matched TCGA cohort. The significance levels of the enrichment were presented as adjusted P-values corrected from the raw P-values based on the Benjamini-Hochberg procedure. The GSEA was performed by the R package fgsea (v1.20.0)[54].

For both approaches, cell lines were ranked based on Spearman's ($\rho$) and NES from high to low, respectively. Then, the two ranking lists were combined, and cell lines were reordered according to their average rank (Supplementary Data 5).

In addition, we also analyzed cell line similarity to the TCGA cohorts at the disease stage and molecular subtype levels. For this, the AJCC (American Joint Committee on Cancer) pathologic stages and molecular subtypes of the analyzed TCGA samples were retrieved using the R package TCGAbiolinks. Specifically, molecular subtypes were retrieved using the package function PanCancerAtlas_subtypes(), and the column "Subtype_Selected" was selected as the final categorization. For each TCGA cohort, gene expression nTPM was averaged

per stage, respectively. Then, Spearman's correlation was calculated between stages and cohort-matched cell lines. Considering the limited number of pathologic stages per cohort, here we selected genes having fourfold higher expression in one stage than any other stage within a TCGA cohort as the stage signature, and calculated the enrichment of the stage signature in the cohort-matched cell lines using the same approach to the GSEA of TCGA disease signature in all cancer cell lines. Finally, correlation-based and GSEA-based ranking lists were combined to obtain the final ranking list for cell line prioritization for the disease stage (Supplementary Data 6). The molecular subtype-level analysis was performed in the same manner (Supplementary Data 7).

### Pathway and cytokine analysis

A total of 14 cancer-related pathways activity for all analyzed cell lines were inferred based on the PROGENy, a package that relies on biological data mining of publicly available data to obtain cancer-related pathway-responsive genes for human and mouse[22]. For this, read counts for HPA and CCLE cell lines quantified by Kallisto were re-analyzed without filtering out the non-protein-coding genes to ensure a broadened coverage of cancer pathway-responsive genes. Specifically, read counts were aggregated by tximport, and 33 common cell lines between HPA and CCLE were combined by summing up the read counts, resulting in 36,498 genes for 1,055 unique cell lines (1,019 from CCLE plus 69 from HPA minus 33 common cell lines) for pathway analysis. Then, the read counts were normalized by DESeq2 (v1.34.0)[55] with respect to the size factor of each cell line and were further transformed by variance stabilizing transformation into $\log_2$ space. To calculate the relative pathway's activities across all cell lines, the normalized values were centered by subtracting the mean value per gene. Then, the R package decoupleR (v2.0.1)[56] was used to calculate the relative pathway's activities based on the top 100 signature genes per pathway obtained from the R package progeny (v1.16.0) as suggested by the original publications[22,57]. By default, the decoupleR was executed using the top performer methods benchmarked (i.e., mlm for multivariate linear model, ulm for univariate linear model, and wsum for weighted sum)[56], and the results were integrated to obtain a consensus score presented as z-score to represent the pathway activity. Here, a consensus z-score above 1 or below −1 was considered significant, resulting in 4,352 (29%) significant pathways among the total 14,770 (14*1055) calculated pathways (Supplementary Data 8).

Similarly, the activity of the 43 CytoSig cytokines was inferred based on the gene expression profile of the 1,055 unique cell lines by the package CytoSig (v0.0.2)[23]. Gene expression data were processed in the same way as for PROGENy analysis. Also, DESeq2 normalized expression values were centered per gene as suggested[23]. The CytoSig program was executed with 10,000 permutations, and the results were presented as z-scores to represent the relative cytokine activities, with a p-value < 0.05 as significant. In total, 25,391 (56%) of the total 45,365 (43*1055) cytokine activities were significant (Supplementary Data 8).

In addition, PROGENy and CytoSig analyses were applied to an independent cell line RNA-seq dataset from Genentech Inc.[26] (n = 610 cell lines) as well as the TCGA dataset analyzed in this study. For this, the expression of the Genentech and TCGA datasets was independently processed and analyzed for the pathway and cytokine activities in the same way as for the HPA and CCLE cell lines. After averaging the gene expression nTPM of the TCGA dataset per cohort, the pathway and cytokine activities were inferred for TCGA at the cohort level. The mean squared error was calculated based on the pathway and cytokine activities, respectively, between the HPA + CCLE cell lines and the Genentech cell lines as well as the matched TCGA cohorts. Further, similar to the transcriptomics-based cell line prioritization, HPA and CCLE cell lines were prioritized for the corresponding TCGA cohorts based on the combined ranking list of pathway and cytokine similarity represented by mean squared error (the lower, the more similar) (Supplementary Data 9).

## Sample visualization

Dimensionality reduction was performed on the nTPM expression data, the PROGENy pathway activity, and the CytoSig cytokine activity for sample visualization. First, we applied principal component analysis (PCA) using the R package pcaMethods (v1.86.0)[58] on the z-normalized values, with the cut-off of the total ratio of variance that is being explained by the principal components (PCs) no less than 0.8 for the number of selected PCs. After PCA, the Uniform Manifold Approximation and Projection (UMAP)[14] analysis was performed by the R package uwot (v0.1.11) to compress the PCs into two dimensions for visualization.

## Statistical analysis

Wilcoxon rank-sum test (after Shapiro-Wilk test for data normality) was used to evaluate the statistical difference between groups. Statistical tests were performed in R (v4.1.2).

## Reporting summary

Further information on research design is available in the Nature Portfolio Reporting Summary linked to this article.

## Data availability

The CCLE publicly available RNA-seq data used in this study are available in the Sequence Read Archive (SRA) database under accession code PRJNA523380[6]. The RNA-seq data of the HPA cell lines generated for this study have been deposited in the Gene Expression Omnibus database under accession code GSE240542. The Genentech RNA-seq data is available under restricted access, access can be obtained by request from the European Genome-phenome Archive (EGA) under accession number EGAS00001000610[26]. The TCGA data are publicly available at https://portal.gdc.cancer.gov/. The processed gene expression data for CCLE and HPA cell lines are available to download on the Human Protein Atlas resource download page [https://v22.proteinatlas.org/about/download]. The remaining data are available within the Article, Supplementary Information or Source Data file. Source data are provided with this paper.

## Code availability

The scripts required to reproduce the results presented in this paper are available in the GitHub repository (https://github.com/jha14/HPA_cell_line)[59].

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

## Acknowledgements

We thank the entire staff of the Human Protein Atlas program and the Science for Life Laboratory (SciLifeLab) for their valuable contributions. This work was supported by WCPR grant from Knut and Alice Wallenberg Foundation, the SciLifeLab & Wallenberg Data Driven Life Science Program (KAW 2020.0239, H.J., C.Z., M.U., A.M.), and the Swedish Research Council Grant 2020-06175 (M.U.). The results shown here are in part based upon data generated by the TCGA Research Network: https://www.cancer.gov/tcga.

## Author contributions

H.J., C.Z., and M.U. conceived and designed the study. H.J., C.Z., and M.Z. collected and curated the data. H.J. and C.Z. performed the data analysis. H.J. and C.Z. drafted the original manuscript. H.J., C.Z., M.K., M.S., M.Y., X.S., X.L., H.Y., M.U., and A.M. revised the manuscript. H.J., C.Z., M.Z., and K.F. visualized the results and created the database portal. C.Z., H.T., L.F., M.U., and A.M. supervised the study. All authors read and approved the final manuscript.

## Funding

## Competing interests

The authors declare no competing interests.
