## [Peer Review File · Nature Communications]

Reviewers' Comments:

Reviewer #1:

Remarks to the Author:

In this manuscript, Jin and colleagues report the integrative analysis of transcriptomics data from cell lines publicly available in the Cancer Cell Line Encyclopedia (CCLE) and the human protein atlas (HPA). The expression data was calculated as nTMP (normalized transcripts per million). The CCLE data was incorporated into the HPA data base without batch correction as initial analysis confirmed feasibility of this approach. Cluster analysis revealed that the majority of cancer cell lines grouped according to the cancer type. A comparison with the gene expression data of the TCGA cancer data base confirmed that a high number of transcripts overlapped but also several were specific for the cancer cell line or the cancer. The comparison of different cell lines to the TCGA data set aimed at providing information on which cell line best represents the corresponding cancer type. Finally, the software packages PROGENy and CytoSig, were utilized to predict based on the expression data pathway activation and cytokine expression of the cell lines.

The work performed by the authors is a valuable asset for the community, as it increases the transcriptomics data sets accessible through the HPA data base and provides some very general insights on how close certain cell lines are to the respective cancer. However, the manuscript does not provide major scientific findings beyond this point. The study was entirely performed with publicly available transcriptomics data and no validation experiments were performed.

Major comments:

1. The title „Is cell line a good model for human cancer?“ is grammatically incorrect and overstates the presented findings as the analyses were only performed on gene expression data of cell lines and compared to gene expression in corresponding bulk tumor tissue. Therefore, the title, albeit catchy, needs to be tuned down.

2. The statement that drug candidates fail because the wrong cell line model was used (Line 41 onwards) is an oversimplification. Tests are usually performed with more than one cell line and drug candidates rather fail because of metabolic alterations and reduced biological availability in humans. These aspects should be considered.

3. To explore to which extent gene expression in cell lines resembles the transcription profiles of the corresponding cancer, a comparison with data of the TCGA data base was performed. It is not considered that the TCGA data sets represent results for bulk tumor tissue and thus comprise also effects of the tumor-microenvironment such as infiltration of immune cells, an aspect that is clearly not present in the context of cell lines. Therefore, the results (lines 157-162) reporting the association of high specificity genes in cancer cell lines with immune-related responses should be reconsidered.

4. For the discrete classification of gene specificity (lines 192-195) the TAU score is introduced. However, the overlap of genes varies greatly and if performed group-wise (enriched, enhanced, etc.) the TAU correlation would probably vanish. As the TAU score is only introduced in one paragraph referring to two supplementary panels, it would be advantageous to remove the entire paragraph.

5. Cancer cell line enriched genes are compared to the genes enriched in the corresponding tissue and the TCGA cancers (lines 219-220). It is not surprising that e.g. hepatoma cell lines are indeed of hepatic origin. This has already been previously confirmed and thus is not novel.

6. Although spearman correlation indicates a correlation between cell line type and TCGA data, it would be important to employ a stronger statistical analysis (lines 238-240), since common expression of more than 5000 genes is observed and almost all presented spearman-correlation values are between 0.7 and 0.85.

7. The TCGA data offers not only information on the primary site for the acquired data, but also on the disease type (eg. adenoma, neoplasms, glioma). If the intention of the authors is to provide information on the suitability of specific cancer cell lines for cancer studies, it would be beneficial to rather focus on the combination of primary site and disease type and repeat the performed analysis. For example, the LUAD (primary site lung) data set includes the following disease types: adenoma and adenocarcinoma, acinar cell neoplasm and mucus and serous neoplasm (other examples are PAAD, BRCA, COAD) (lines 258-261). The novelty of the analysis is limited as other studies have already covered this in much more depth (e.g. Nwosu et al., J Exp Clin Cancer Res. 2018 37(1):211). Further, it is a major oversimplification to imply that the expression of albumin is a good marker of a cell line to represent liver cancer.

8. The authors suggest that a hallmark of TCGA-LIHC (LIHC is an unusual abbreviation and rather

HCC standing for hepatocellular carcinoma should be used) is lost in SNU-398. This conclusion appears rather arbitrary as there are no mechanisms evident that might be influenced by the "lowly-expressed" genes in this cell line.

9. The analysis of cytokine expression and pathway activation primarily confirms the employed software packages. It should be pointed out to which extent novel findings were obtained.

Minor comments:

1. The authors should discuss the advantage of reporting the data as pTMP, rather than FPTM or RPTM.
2. The UMAP projection shown in Fig. 1C should have axis labels, ticks and tick labels.

Reviewer #2:

Remarks to the Author:

In the current manuscript, Jin et al. present a transcriptome-driven selection process by which cell lines can be ranked according to their biological relevance to a given disease state such as a particular cancer type. In brief, the authors attempted to compare the transcriptomic landscape of more than 1,000 human cell lines representing 27 cancer types with the corresponding profiles in various tissues, organs, single cell types, and diverse tumor states. In addition, the authors investigated cancer-related pathways and cytokine gene expression of the cell lines. I highly appreciate that the authors made the effort to make the data available in the open-access cell line section of the Human Protein Atlas.

I find the focus of the study emphasizing the investigation of the relevance of a particular cell line for a given disease state through genome-wide gene expression analysis. Although the focus of the study is interesting and certainly has the potential to attract the attention of the scientific community, the study lacks validation using data from independent cohorts. Furthermore, the notion of comparing the cell lines directly with tissue counterparts is problematic without accounting for cellular heterogeneity and the immune fraction that is present in the tissue samples. Moreover, the authors did not consider the different clinical attributes such as cancer stages and other molecular (mutational data) and clinicopathological criteria of the TCGA tumors.

Major Points:

1. Comparing the transcriptome between cell lines and tumor tissue counterparts for a particular cancer type is problematic simply because tumor tissue contains heterogeneous cell types specifically infiltrating immune cells. Therefore, it is important to take into account the immune fraction and tumor purity of the tumor tissue. I recommend the authors to check for tumor purity and estimate infiltrating immune fractions for the TCGA tumor samples. Lower tumor purity and higher immune fractions can eventually explain the subsequent lower correlation between a particular cell-line and its corresponding tumor tissue counterpart. Authors can take advantage of the ESTIMATE package for this purpose. In addition, the authors should check the ploidy of the TCGA samples and take copy number variations into account when comparing with cell lines.

Following the same notion, it would be interesting to see the concordance of cell-lines and pre-stratified TCGA samples based on different clinical and molecular categories. For example, it is possible to extract the transcriptomics-based molecular subtypes of TCGA samples as well as clinical stages. Integrating the molecular subtype and clinical data may reveal some subtype- and stage-specificity of the cell lines which would be more interesting from a biological point of view.

2. I did not find any comparison of the mutational status of the cell-lines and TCGA samples. Since the mutational landscape is an important determinant of the relevance of cell lines to a particular cancer type, it is pertinent to instigate the mutational status of the CCL /HPA cell lines, along with the TCGA samples, and subsequently interrogate whether the specific mutational landscape has the ability to influence the concordance between cell-line and corresponding tumor sample.

3. The concordance between a cell line and corresponding tumor tissue is based on the transcriptome-derived correlation and enrichment analysis. The authors did not provide any validation for their claim that the correlation-based and enrichment-based approaches are efficient

to predict how similar a cell line is to its corresponding cancer type. The lack of experimental validations to support the claims weakens the credibility of such approaches. Authors should consider validating the approaches by performing additional experiments and/or by using additional data from independent cohorts.

4. The authors used single-cell RNA-seq data and compared the "tau" scores between cell lines and single cell types (Supp Figure 4G). Here the annotation or cell type identity of the single cells would be helpful to explain the results in a biologically meaningful way. By considering the cell type annotations of the single cells it is possible to identify the malignant, non-malignant and immune cells. The presence of non-malignant and immune cells may lead to the considerable imbalance toward single-cell types that is obvious in Supp Figure 4G.

5. The section presenting the analysis of cancer-related pathways and cytokine activity in human cell lines is somewhat speculative. Here again, the authors should provide additional experimental evidence in support of their claim that specific cancer-related pathways and cytokines are upregulated in certain cancer cell lines. Alternatively, authors should consider presenting validation by analyzing data from an independent cohort (cell-line RNA-seq data that is not part of HPA and CCLE). Also, just mere up-regulation of certain pathways and cytokines does not fully justify the biological context. To enable the readers to interpret the results in a biologically meaningful manner, it is necessary to analyze these pathways and cytokines in TCGA tissue samples as well and try to assess the concordance of these pathways and cytokines in the context of the tissue microenvironment. The discordant pathways and cytokines between cell-lines and corresponding TCGA samples can either be cell-line specific (therefore not activated in TCGA samples) or related to the microenvironment (absent in the cell lines).

6. I agree that the author attempted to answer the question as stated in the title. However, the methods applied by the authors only involve transcriptomics analysis and lack any phenotypic and histologic interrogations. Therefore, it is necessary to state the specific method (eg: Transcriptomics) explicitly in the title that was used in the study to address the question. I would recommend changing the title to a more conventional one highlighting the transcriptomics analysis.

Minor points:

1. Line: 145-148: How the enrichment was done?
2. Line: 118-119: What is number of common genes expressed in all CCLE and HPA cell lines?
3. Line: 124: Include the reference of GSOA.
4. 143-144: What are the resources? For clarity, it is better to explicitly state the resources.
5. Line: 347-348: Is any reference available showing that PC-3 is indeed not suitable for androgen suppression therapy?

Reviewer #3:

Remarks to the Author:

The authors investigate the concordance between a large panel of cancer cell lines and tumours from a transcriptional point-of-view. This is a long-posed question that has been investigated in previous studies in the context of individual cancer types or in a pan cancer manner. To this purpose, the authors assess the transcriptional specificity of measured genes in cell lines and primary tumours as well as single-cell data from healthy tissues. In this way the authors report a reasonable enrichment of cell line cancer-specific genes which is concordant to what observed in TCGA primary tumour samples. They also performed an interesting analysis comparing metastatic cell lines expression with primary tumours of the same tissue or the same site. Additionally, they merged two gene expression based criteria to find the top 5 cell lines representing a cancer type and finally investigated pathway and cytokine activity scores at the cell line level to inform on phenotypic features that could be used in in-vitro drug screenings.

Overall, this is an interesting work tackling an important problem. The manuscript is well-

structured and provides a valuable tool for the selection of cell-line most transcriptionally similar to primary tumours.

However, this topic has already been extensively studied and several tools are now available pursuing aims similar to those of motivating this study. The authors fail to highlight the innovative aspects of the proposed method/analysis. Particularly, a systematic comparison with previous results from tools for the selection of good-quality cell lines should be compulsorily added important.

In addition, the following points should be addressed.

- The title is a question that is never answered throughout the text. The authors should draw some conclusive remarks in the form of answers to the title. In addition, it reads odd, I would suggest to change it in: "Are cell lines good models of primary tumours?" an alternative would be to opt for a more descriptive title that is reflective of the performed analyses and results. Something along the lines "Systematic transcriptional analysis of cell lines ... etc"

- The authors claim that "a comprehensive comparison of gene expression between cell lines, human tissues, tumours and cells is still lacking". They should mention that such comprehensive comparison has already been performed, at least between cell lines and tumours in PMID: 33397959 and put more emphasis on the difference between this work and their analysis, which is original because is extended to normal tissues and individual cells. This would also help to highlight the novelty of their work generally.

- In PMID: 33397959, the authors report a 'central cluster' cell lines presenting a more mesenchymal and undifferentiated transcriptional state. I can't see any reference to this finding. It is important that the authors mention this and characterise in depth these cell lines.

- Other important previous studies similar to what presented in this manuscript should be cited and briefly described, again highlighting differences and commonalities with what the authors present here: particularly Sinha et al (2021) (TumorComparer), Zhang & Kschischo (2021) (MFmap) and Salvadores et al (2020) (HyperTracker)

- The lack of batch effects between the HPA and CCLE transcriptional datasets comes out clearly from supplementary figure 2, although it should be quantified. Particularly, the authors claim that cell lines in common across the two datasets lie 'tightly' close to each other in the integrated UMAP space. This qualitative claim is not acceptable and the authors should prove a more robust estimation such as for example percentage of cell lines in one dataset that have as 1st, 2nd, 3rd, ... closest neighbor their counterpart in the other dataset. The dendrogram in Supplementary figure 3 is not enough. In addition, in that figure, most of the cell lines shared across the two datasets have their two labels superimposed, creating an annoying anaglyph-3D-like effect. The authors should work on an alternative visualisation.

- Similar considerations apply to the qualitative observations that the authors made on cell lines from certain cancer type being overall more similar/dissimilar to the others, based on figure 1c. These overall similarities should be quantified, for example computing the ratio between inter/intra-cluster variance for each cancer type.

- Again, the authors claim: "more than half of the cell lines were grouped by cancer types, suggesting that cell line preserve the cancer phenotype at the transcriptional level". This claim is not supported by any quantitative measurement applied to the clusters in figure 1c, their ability to correctly classify a single cancer type, nor statistical enrichment of cancer types across clusters, ROC indicators. Deriving such biological conclusions just by eyeballing a uMAP plot is not acceptable.

- The authors reference Table 1 to explain how they have explored gene expression distributions and specificity across cancer type. Unfortunately this table is not clear at all. They should briefly but clearly describe their analysis in the main text.

- What is the point of highlighting transcriptional differences across CCLE and HPA? (figure 2a)

- The authors draw conclusions from comparing housekeeping genes and 'essential' genes but referencing not very recent works based on CRISPR-screening very few cell lines. Much more comprehensive and more recent CRISPR/shRNA screens have been performed on thousands of cancer cell lines (potentially covering most of the CCLE cell lines) and should be used for these comparisons (for example, the works/datasets presented in: PMID: 28753430, PMID: 30971826, PMID: 33712601).
- The choice of 'cell line cancer type' to refer to transcriptional profiles resulting from cell line aggregation on a cancer type level is quite unfortunate and confusing. The authors should opt of an alternative, even an acronym would be better.
- The analysis summarised in figure 2d is well conceived but, again, results are summarised in a qualitative rather than quantitative way and this should be amended. What is the gene grouping similarity across categories between Cell line cancers and TCGA cancers? There are suitable metrics to assess this.
- The authors find that high specific genes identified in cell line cancer types but not in tumours are enriched for immune-related biological processes. This is very puzzling, as cell line are lacking immune components. The inverse associations would have been more plausible. The authors should comment and justify this.
- cell lines are grouped by their primary disease, but it was shown in previous studies (e.g. Salvadores et al (2020)) that some lines are mislabelled or transdifferentiated. I suggest looking at the mentioned publication and add a comment on it.
- in the comparison with TCGA, it would make more sense to use the common set of cancer type. Indeed, the different sets of high-specific genes arise from TCGA lack of blood cancer cohorts (line 176).
- From their analysis the authors conclude that gene specificity is similar. I understand it is relevant when looking at the same cancer type, but I do not get why it is useful when across data-set they do not even use the same set of tissue type.
- correlation methods to match cell lines and tumours have been widely used (many reviewed in PMID: 35822563). Since the raking problem from transcriptome was also previously addressed, it would be interesting to see a comparison with previous studies (e.g. Yu et al (2019) ranking results).
- The pathway and cytokine based analysis has been vey nicely performed and presented in a set of effective visualisations. It would be interesting to see how progeny and cytosign features are actually distributed in TCGA and compared cancer specific scores with cell lines (similar to what was done in paragraph 4 for spearman correlation), would be the top 5 cell lines also highly concordant from this new features?
- Lines 355-357: I think this is overstated, previous studies have proposed a subset of cell lines as "good" models using gene expression data. For example, Peng et al (2021) (CancerCellNet), Yu et al (2019) (CompHealth) , Salvadores et al (2020). I would stress instead the advantages of the proposed study compared to the previous one

Other minor points:

- While referencing previous works aiming at identifying new therapeutic targets based on functional/chemo-genomics screens of large panels of immortalised human cell lines, the following works could be cited: PMID: 28753430, PMID: 30971826
- Large cell line-based Multi-omic datasets and their use to identify therapeutic biomarkers could also be mentioned. On this regard, the following works might be cited: PMID: 23180760, PMID:

27397505, PMID: 22460902

- Fig. 2e is never references in the main text

Reviewer comments

Reviewer #1 - Systems biology, transcriptomics - (Remarks to the Author):

In this manuscript, Jin and colleagues report the integrative analysis of transcriptomics data from cell lines publicly available in the Cancer Cell Line Encyclopedia (CCLE) and the human protein atlas (HPA). The expression data was calculated as nTMP (normalized transcripts per million). The CCLE data was incorporated into the HPA data base without batch correction as initial analysis confirmed feasibility of this approach. Cluster analysis revealed that the majority of cancer cell lines grouped according to the cancer type. A comparison with the gene expression data of the TCGA cancer data base confirmed that a high number of transcripts overlapped but also several were specific for the cancer cell line or the cancer. The comparison of different cell lines to the TCGA data set aimed at providing information on which cell line best represents the corresponding cancer type. Finally, the software packages PROGENy and CytoSig, were utilized to predict based on the expression data pathway activation and cytokine expression of the cell lines.

The work performed by the authors is a valuable asset for the community, as it increases the transcriptomics data sets accessible through the HPA data base and provides some very general insights on how close certain cell lines are to the respective cancer. However, the manuscript does not provide major scientific findings beyond this point. The study was entirely performed with publicly available transcriptomics data and no validation experiments were performed.

We thank the reviewer for the positive comments. Based on the suggestions, we have thoroughly revised the manuscript to improve the scientific content. Details of the point-by-point response can be found below.

Major comments:

1. The title „Is cell line a good model for human cancer?“ is grammatically incorrect and overstates the presented findings as the analyses were only performed on gene expression data of cell lines and compared to gene expression in corresponding bulk tumor tissue. Therefore, the title, albeit catchy, needs to be tuned down.

Based on the suggestions also from reviewer #2 and #3 (please see comment 6 from R2 and comment 1 from R3), we have changed the title to “Systematic transcriptional analysis of human cell lines for gene expression landscape and tumor representation”.

2. The statement that drug candidates fail because the wrong cell line model was used (Line 41 onwards) is an oversimplification. Tests are usually performed with more than one cell line and drug candidates rather fail because of metabolic alterations and reduced biological availability in humans. These aspects should be considered.

We have rephrased the relevant sentence and added a reference regarding the metabolic alteration of liver cancer cell line¹ to highlight the aspects raised by the reviewer. See page 3, line 42-44:

“Due to metabolic alterations in cell line models (<https://doi.org/10.1186/s13046-018-0872-6>) and reduced biological availability in humans, drug candidates may still have a high chance of failing in the in vivo studies and clinical trials, even though they are successfully tested in vitro based on more than one cell line.”

3. To explore to which extent gene expression in cell lines resembles the transcription profiles of the corresponding cancer, a comparison with data of the TCGA data base was performed. It is not considered that the TCGA data sets represent results for bulk tumor tissue and thus comprise also effects of the tumor-microenvironment such as infiltration of immune cells, an aspect that is clearly not present in the context of cell lines. Therefore, the results (lines 157-162) reporting the association of high specificity genes in cancer cell lines with immune-related responses should be reconsidered.

We thank the reviewer for pointing out this critical issue, and we agree that the TCGA dataset represents transcriptional profiling of bulk tumors in which transcripts from other cells (mostly immune cells and stromal cells) than cancer cells were also sequenced. This aspect has been seriously reconsidered in the revised

manuscript to better align the cancer cell lines with bulk tumors. Specifically, to address this, we have made the following analyses:

1. The transcriptomics data of 26 TCGA cohorts (including blood cancer cohort LAML) have been reanalyzed. Compared to the original manuscript which focused on 21 TCGA cohorts, CHOL (bile duct cancer/cholangiocarcinoma), ESCA (esophageal cancer), LAML (leukemia), LGG (brain cancer), and SARC (sarcoma) have been added for gene expression landscape analysis, cell line prioritization and PROGENy/CytoSig pathway and cytokine analyses. The cancer types between the cell line and TCGA datasets have been aligned with the best efforts.
2. For TCGA data, tumor purity was taken into account. We use ESTIMATE² to calculate tumor purity and remove TCGA samples with high tumor infiltration (tumor purity score < 0.7; see below) to reduce the effect of stromal and infiltrated immune cells from bulk tumors when compared with cell line transcriptomics. In total, 6,082 TCGA samples with high tumor purity were analyzed.

Based on these efforts, the reliability of the results presented in Figure 2d and others has been significantly improved, and the major inconsistency of gene expression specificity between the cell line dataset and TCGA dataset (especially the presence of immune-related responses in cell lines) was overcome. Therefore, we removed the original Figure S4a-c and the corresponding text from the manuscript to improve clarity and readability.

4. For the discrete classification of gene specificity (lines 192-195) the TAU score is introduced. However, the overlap of genes varies greatly and if performed group-wise (enriched, enhanced, etc.) the TAU correlation would probably vanish. As the TAU score is only introduced in one paragraph referring to two supplementary panels, it would be advantageous to remove the entire paragraph.

We agree. Following the suggestions, we have removed the TAU score paragraph and the corresponding figures (Figure S4e-g).

5. Cancer cell line enriched genes are compared to the genes enriched in the corresponding tissue and the TCGA cancers (lines 219-220). It is not surprising that e.g. hepatoma cell lines are indeed of hepatic origin. This has already been previously confirmed and thus is not novel.

We agree that genes enriched in liver cancer cell lines are expected to have significant overrepresentation in the genes enriched in liver tissue, TCGA-LIHC (liver cancer) cohort, and hepatocyte (single-cell type). Although these findings presented in Figure 3a-b are not out of expectation, here we performed this overrepresentation analysis and presented the two figures aiming at demonstrating the consistency of gene expression specificity between cancer cell lines and the corresponding tissues, tumors, and cell types. In this part, tissue, tumors, and single-cell types were all independently analyzed and then compared with cell line data, providing an unbiased comparison of gene expression specificity and landscape between cell

lines and the corresponding tissues, tumors, and cell types. To better reflect our aim, we have rephrased the sentences on page 6, line 192-195.

“After examination, we found that more than half of the enriched genes in liver cancer cell lines were also enriched in liver tissue, TCGA-LIHC cohort, and hepatocytes (Fig. 3b), suggesting a high concordance of enriched genes between liver cancer cell lines and the corresponding tissue, tumor, and cell type.”

6. Although spearman correlation indicates a correlation between cell line type and TCGA data, it would be important to employ as stronger statistical analysis (lines 238-240), since common expression of more than 5000 genes is observed and almost all presented spearman-correlation values are between 0.7 and 0.85.

Following the reviewer’s comment, we realized that the heatmap presented in the original manuscript did not fully distinguish the different levels of correlation between cell line type and TCGA cohorts. In the revised manuscript, with the inclusion of more TCGA cohorts (especially the blood cancer cohort LAML), correlations between cell line diseases and TCGA cohorts are more distinguishable, ranging from 0.655 to 0.852. In addition, we changed the color palette for the heatmap to provide a more eye-comfortable gradient thus high correlations are further highlighted. See below left for the original figure and right for the revised figure.

While a previous study (Yu et al.; TCGA-110-CL³) used the 5,000 most variable genes for correlation analysis (see Figure 1c in <https://www.nature.com/articles/s41467-019-11415-2>), here we argue that using a subset of all protein-coding genes may not reflect the full transcriptome-wise information. Compared with the previous study, a transcriptome-wise analysis based on more than 20,000 protein-coding genes is the advantage of this study.

7. The TCGA data offers not only information on the primary site for the acquired data, but also on the disease type (eg. adenoma, neoplasms, glioma). If the intention of the authors is to provide information on the suitability of specific cancer cell lines for cancer studies, it would be beneficial to rather focus on the combination of primary site and disease type and repeat the performed analysis. For example, the LUAD (primary site lung) data set includes the following disease types: adenoma and adenocarcinoma, acinar cell neoplasm and mucus and serous neoplasm (other examples are PAAD, BRCA, COAD) (lines 258-261). The novelty of the analysis is limited as other studies have already covered this in much more depth (e.g.

Nwosu et al., J Exp Clin Cancer Res. 2018 37(1):211). Further, it is a major oversimplification to imply that the expression of albumin is a good marker of a cell line to represent liver cancer.

This is an excellent suggestion. Following this notion, we have repeated the performed analysis to prioritize cancer cell lines to the corresponding TCGA cohorts at AJCC pathologic stage level and molecular subtype level. For this, clinical information and TCGA molecular subtypes were retrieved by the R package TCGAbiolinks.

Since our cell line prioritization algorithm involves two aspects: transcriptome-wise correlation and GSEA of disease signature, for each TCGA cohort, we calculated cell line similarity to TCGA cohort stages and subtypes based on the following steps:

1. The gene expression nTPM of TCGA samples was averaged by AJCC pathologic stages or molecular subtypes.
2. The correlations between cell lines and TCGA cohort stages/subtypes were calculated based on all available protein-coding genes.
3. Stage- and subtype-specific signature was defined as the genes having four-fold higher expression than any other stage/subtype within one TCGA cohort.
4. Calculating the enrichment of the stage-/subtype-specific signature in each of the cohort-matched cell lines.

Then, we compared the subtype level cell line ranking list to a previous study (TCGA-110-CL³) for two TCGA cohorts BRCA and HNSC where the same subtype categorization was used. We found the correlations between our ranking lists and the TCGA-110-CL ranking list for subtype analysis were generally high (> 0.6, all significant, see Supplementary Fig. 6b), demonstrating the validity of our method for cell line selection for subtypes and stages.

As a showcase in this study, we focused on the TCGA-LIHC cohort and examined prioritized cell lines (especially Huh-7 and Hep-G2, which ranked 1st and 2nd for the cohort) for different pathologic stages and molecular subtypes. We found that for most of the stages and subtypes, Huh-7 outperformed Hep-G2. But when focusing on pathologic stage II, Hep-G2 was ranked higher than Huh-7 (2nd and 6th, respectively). These results suggested that the Huh-7 can be a good candidate across all tumor stages and subtypes, whereas for specific conditions, other cell lines could be good options. We have added the stage and subtype analyses on page 8, line 280 to page 9, line 291, and Supplementary Table 6-7.

Rather than showing ALB as a good marker of liver cancer cell lines, here we presented the expression of ALB in liver cancer cell lines to indicate that some important tissue/tumor characteristics may be underrepresented in some of the liver cancer cell lines. Despite this having already been demonstrated using GSEA, a barplot showing the expression of ALB would be a straightforward visualization to represent the enrichment of the TCGA disease signature in cell lines. Considering that, we tend to keep this result in the manuscript, and added a note (page 8, line 261-264) to highlight our aims:

“For instance, the Albumin (ALB) gene – one of the TCGA-LIHC signature genes, which is also a hallmark gene of liver functions, was highly expressed in the top-ranked liver cancer cell lines (Fig. 5b), distinguishing prioritized cell lines from lowly ranked cell lines.”

8. The authors suggest that a hallmark of TCGA-LIHC (LIHC is an unusual abbreviation and rather HCC standing for hepatocellular carcinoma should be used) is lost in SNU-398. This conclusion appears rather arbitrary as there are no mechanisms evident that might be influenced by the “lowly-expressed” genes in this cell line.

We thank the reviewer for this thoughtful suggestion. To demonstrate the associated biological functions with the hallmark (signature) of the TCGA-LIHC cohort, we performed gene set overrepresentation analysis and found these LIHC signature genes are associated with metabolic functions in liver (Figure 5c), meanwhile showing relatively low expression in lowly-ranked liver cancer cell lines compared to highly-ranked cell lines. These results suggested that the cell lines with relatively low expression of the disease signature may have underexpression of basic tissue functions and are thus less representative of the corresponding tissue/tumor. This analysis has been added on page 8, line 264-266.

9. The analysis of cytokine expression and pathway activation primarily confirms the employed software packages. It should be pointed out to which extent novel findings were obtained.

Here we provided such information for more than 1,000 cell lines to facilitate researchers' selection of the appropriate cell lines for basic research, clinical and translational purposes. The meaning of providing such information was demonstrated in Figure 7 and discussed in the third paragraph of the discussion section. To further highlight the novelty of the pathway/cytokine analysis, in the revised manuscript, we performed extra analyses to make this part more technically robust and scientifically meaningful.

1. First, to validate the cancer-related pathways and cytokines activity, we calculated pathway/cytokine activities on an independent cancer cell line RNA-seq dataset from Genentech, Inc.⁴ (n = 610, with 461 overlapping with the cell lines analyzed in the manuscript) to validate the reliability of the results presented in the manuscript. Based on the mean square error (MSE) comparing pathway/cytokine activities between each pair of the common cell lines from HPA+CCLE and Genentech, results showed that the identical cell lines from different datasets have very similar pathway/cytokine profiles. The results have been added on page 9, line 314-319.
2. Second, the pathway/cytokine activities were also estimated for TCGA cohorts. Based on the PROGENy/CytoSig activities, we again calculated the similarity between cancer cell lines and the corresponding TCGA cohorts based on the mean square error (MSE), and prioritized top-5 (including tied) cell lines based on the combined ranking of PROGENy and CytoSig activities. We found that 31 of the total 112 pathway- and cytokine-based selected cell lines can be found in the transcriptomics-based selected cell lines (Figure 4) (hypergeometric testing $p = 4.14E-07$), suggesting a significant overlap of the prioritized cell lines between pathway/cytokine analysis and transcriptomics analysis. Of note, the PROGENy and CytoSig were designed to investigate the relative activities of the analyzed pathways and cytokines in the samples within one dataset^{5,6}. To ensure a fair and unbiased comparison, the HPA+CCLE, Genentech cell line datasets, and TCGA datasets were processed and analyzed separately, with different standard levels to represent the relative activities of the cell lines/tumors in each dataset (this is suggested by the authors in their original publications^{5,6}). This may exaggerate the differences in the pathway/cytokine activities between these three datasets. Even though, the significant overlap of the selected cell lines demonstrated the usage of the two computational tools and highlighted the value of presenting pathway and cytokine activities for the analyzed cell lines. This analysis has been added on page 9, line 319-325.

While these results are meaningful, we did not further dig into the pathway-/cytokine-based cell line selection, but used it as a validation for the two computational tools. The main reason is that the 14 pathways and 43 cytokines, despite being informative, may not cover the full transcriptomics information. For example, in the revised Figure 5 we demonstrated that some fundamental metabolic processes in liver tissue were underexpressed in lowly-ranked liver cancer cell lines. This key information for cell line prioritization may not be well-reflected by pathway and cytokine analysis.

Minor comments:

1. The authors should discuss the advantage of reporting the data as pTMP, rather than FPTM or RPTM.

The aim of using TMM⁷ normalized gene expression (same pipeline as the other datasets analyzed in HPA database) is to allow the direct comparison between tissue, cancer, single-cell type, and cell line gene expression under the Human Protein Atlas frame. As reported⁷, TMM was designed to overcome the limitation induced by the sequencing platform when aberrant high expression of a gene occurs, where using TPM with a fixed library size per sample may lead to the dilution of the other genes. We have revised the discussion to highlight the advantage of using nTPM rather than (p)TPM for expression analysis (page 12, line 408-414).

“For all the cell lines analyzed in this study, we processed the gene expression normalization using the same pipeline applied to the other HPA resources, and present the normalized expression nTPM of these cell lines in the HPA database. Based on our previous analysis, the TMM normalized gene expression can minimize the batch effects caused by sampling and different technology platforms, especially when a small

number of genes are very highly expressed, facilitating the sample-wise comparison between gene expression in cell lines and tissues, organs, single cell types, and cancer biopsies.”

2. The UMAP projection shown in Fig. 1C should have axis labels, ticks and tick labels. Following the suggestions, we have added axis labels, ticks, and tick labels. Changes can be found in Figure 1 and Figure S2.

Reviewer #2 - Computational, systems biology, omics - (Remarks to the Author):

In the current manuscript, Jin et al. present a transcriptome-driven selection process by which cell lines can be ranked according to their biological relevance to a given disease state such as a particular cancer type. In brief, the authors attempted to compare the transcriptomic landscape of more than 1,000 human cell lines representing 27 cancer types with the corresponding profiles in various tissues, organs, single cell types, and diverse tumor states. In addition, the authors investigated cancer-related pathways and cytokine gene expression of the cell lines. I highly appreciate that the authors made the effort to make the data available in the open-access cell line section of the Human Protein Atlas.

I find the focus of the study emphasizing the investigation of the relevance of a particular cell line for a given disease state through genome-wide gene expression analysis. Although the focus of the study is interesting and certainly has the potential to attract the attention of the scientific community, the study lacks validation using data from independent cohorts. Furthermore, the notion of comparing the cell lines directly with tissue counterparts is problematic without accounting for cellular heterogeneity and the immune fraction that is present in the tissue samples. Moreover, the authors did not consider the different clinical attributes such as cancer stages and other molecular (mutational data) and clinicopathological criteria of the TCGA tumors.

We thank the reviewer for the thoughtful comments. Based on the comments, we have deeply revised the manuscript to overcome the limitation presented in the original manuscript. Details of the point-by-point response can be found below.

Major Points:

1. Comparing the transcriptome between cell lines and tumor tissue counterparts for a particular cancer type is problematic simply because tumor tissue contains heterogeneous cell types specifically infiltrating immune cells. Therefore, it is important to take into account the immune fraction and tumor purity of the tumor tissue. I recommend the authors to check for tumor purity and estimate infiltrating immune fractions for the TCGA tumor samples. Lower tumor purity and higher immune fractions can eventually explain the subsequent lower correlation between a particular cell-line and its corresponding tumor tissue counterpart. Authors can take advantage of the ESTIMATE package for this purpose. In addition, the authors should check the ploidy of the TCGA samples and take copy number variations into account when comparing with cell lines.

Following the same notion, it would be interesting to see the concordance of cell-lines and pre-stratified TCGA samples based on different clinical and molecular categories. For example, it is possible to extract the transcriptomics-based molecular subtypes of TCGA samples as well as clinical stages. Integrating the molecular subtype and clinical data may reveal some subtype- and stage-specificity of the cell lines which would be more interesting from a biological point of view.

These are valid points. To overcome the pitfall that bulk tumors from the TCGA database also comprise immune cells infiltrated, we use ESTIMATE² to calculate tumor purity and remove TCGA samples with high tumor infiltration (tumor purity score < 0.7; see below) to reduce the effect of stromal and immune cells from bulk tumors when compared with cell line transcriptomics. In total, 6,082 TCGA samples with high tumor purity were analyzed. Results presented in Figure 2d and others have been significantly improved when comparing cancer cell lines with TCGA tumors.

Following the suggestions, we also repeated the performed analysis to prioritize cancer cell lines to the corresponding TCGA cohorts at AJCC pathologic stage level and molecular subtype level. For this, clinical information and TCGA molecular subtypes were retrieved by the R package TCGAbiolinks.

Since our cell line prioritization algorithm involves two aspects: transcriptome-wise correlation and GSEA of disease signature, for each TCGA cohort, we calculated cell line similarity to TCGA cohort stages and subtypes based on the following steps:

1. The gene expression nTPM of TCGA samples was averaged by AJCC pathologic stages or molecular subtypes.
2. The correlations between cell lines and TCGA cohort stages/subtypes were calculated based on all available protein-coding genes.
3. Stage- and subtype-specific signature was defined as the genes having four-fold higher expression than any other stage/subtype within one TCGA cohort.
4. Calculating the enrichment of the stage-/subtype-specific signature in each of the cohort-matched cell lines.

Then, we compared the subtype level cell line ranking list to a previous study (TCGA-110-CL³) for two TCGA cohorts BRCA and HNSC where the same subtype categorization was used. We found the correlations between our ranking lists and the TCGA-110-CL ranking list for subtype analysis were generally high (> 0.6, all significant, see Supplementary Fig. 6b), demonstrating the validity of our method for cell line selection for subtypes and stages.

As a showcase in this study, we focused on the TCGA-LIHC cohort and examined prioritized cell lines (especially Huh-7 and Hep-G2, which ranked 1st and 2nd for the cohort) for different pathologic stages and molecular subtypes. We found that for most of the stages and subtypes, Huh-7 outperformed Hep-G2. But when focusing on pathologic stage II, Hep-G2 was ranked higher than Huh-7 (2nd and 6th, respectively). These results suggested that the Huh-7 can be a good candidate across all tumor stages and subtypes, whereas for specific conditions, other cell lines could be good options. We have added the stage and subtype analyses on page 8, line 280 to page 9, line 291, and Supplementary Table 6-7.

Regarding the copy number variations and ploidy, we have discussed the advantages/disadvantages of including/excluding the CNVs and ploidy below in reply to comment 2.

2. I did not find any comparison of the mutational status of the cell-lines and TCGA samples. Since the mutational landscape is an important determinant of the relevance of cell lines to a particular cancer type, it is pertinent to instigate the mutational status of the CCLE /HPA cell lines, along with the TCGA samples, and subsequently interrogate whether the specific mutational landscape has the ability to influence the concordance between cell-line and corresponding tumor sample.

We agree with the reviewer that the inclusion of the mutational data as well as CNV/ploidy could provide more comprehensive information for cell line prioritization. In this study, the main reason for us to focus on transcriptomics is that, compared to the other omics such as mutation and copy number variation, as the downstream of genetic alterations, transcriptomics is particularly valuable and can reflect the changes in the genetic level. Many studies have reported that genetic alterations are insufficient in predicting drug response, which often rely on transcriptional expression to further characterize the mutation landscape^{8,9}. Indeed, studies showed that these mutational changes could be captured in the expression profiles^{10,11}. Considering that the functional consequences of genetic mutations can be well-captured by gene expression, in this study we restricted our analyses to transcriptomics only, and retitled the manuscript accordingly. Nevertheless, integrating cell line mutation/CNV as well as newly published proteomics¹² and metabolomics¹³ datasets as a multi-omics study will increase the robustness of cell line selection. Indeed, a few studies have utilized multi-omics to study cell line characteristics and to prioritize cell lines for cancer research^{14,15}. As an extension of the current study, this important comment can potentially guide an independent study with a specific focus in the future. For this, we have added a note in the discussion to highlight the future direction (page 12, line 428-431).

“Future works would focus on multi-omics analysis of cell lines using genomics and recently released proteomics, metabolomics and drug response datasets to comprehensively evaluate and expand the main findings in this study.”

3. The concordance between a cell line and corresponding tumor tissue is based on the transcriptome-derived correlation and enrichment analysis. The authors did not provide any validation for their claim that the correlation-based and enrichment-based approaches are efficient to predict how similar a cell line is to its corresponding cancer type. The lack of experimental validations to support the claims weakens the credibility of such approaches. Authors should consider validating the approaches by performing additional experiments and/or by using additional data from independent cohorts.

We acknowledge that experimentally validating the selection of the highly-ranked cell lines provided in this study is challenging, since the collection of these cell lines in lab (even for a single TCGA cohort) is hard to achieve. Meanwhile, there are still no comprehensive and effective evaluation metrics when assessing the global fidelity of cell line models to the corresponding cancer types *in vitro*. Nevertheless, we attempt to address this issue by comparing our results with competing methods from independent studies, i.e., the TCGA-110-CL panel provided by Yu et al.³

After aligning the cell line diseases and TCGA cohorts with Yu et al. study, we found that 65 of the total 114 selected cell lines in this study were presented in the TCGA-110-CL panel ($n = 100$) (hypergeometric testing $p = 1.55E-44$; Figure S6a). This significant overlap supports the validity of the selected cell lines in our study. Of note, Yu et al. study only relies on the correlation based on the 5000 most variable genes, while our analysis is transcriptome-wise based using more than 20,000 protein-coding genes – a significant improvement compared to the previous study. The other advantage of our study is the use of GSEA to estimate the enrichment of the TCGA disease signatures, which have not been considered in the previous study. The validity of GSEA has been demonstrated in the new Figure 5, showing that some liver cancer cell lines have weaker disease features compared to others. Taken together, our 2D (correlation-based and GSEA-based) cell line prioritization method outperforms the previous correlation-based studies.

The results discussed above have been added to the manuscript (page 8, line 250-255; page 11, line 379-382).

“To validate the selected candidates, we compared the results with a previous study using 5,000 most variable genes for correlation-based cell line selection (TCGA-110-CL; $n = 100$ selected cell lines for the same cohorts analyzed in this study). We found that 65 of our selections were also included in the TCGA-110-CL panel, with a p -value of $1.55E-44$ based on hypergeometric testing, underpinning the validity of the selected cell lines by our approach.”

“Especially the adoption of the GSEA, which showed discriminative in estimating fundamental tumor biological processes in cell lines (Figure 5c), has not yet been investigated in the previous correlation-based study.”

4. The authors used single-cell RNA-seq data and compared the “tau” scores between cell lines and single cell types (Supp Figure 4G). Here the annotation or cell type identity of the single cells would be helpful to explain the results in a biologically meaningful way. By considering the cell type annotations of the single cells it is possible to identify the malignant, non-malignant and immune cells. The presence of non-malignant and immune cells may lead to the considerable imbalance toward single-cell types that is obvious in Supp Figure 4G.

Currently, the single-cell type section in the Human Protein Atlas database focuses on healthy tissues and thus does not include malignant cells¹⁶. Our colleagues are expanding the single-cell type section to include tumor cell types. The imbalance toward single-cell types is mainly caused by the number of single-cell types analyzed ($n = 51$), which is higher than the analyzed diseases in the cell line ($n = 27$) and TCGA ($n = 26$ cohorts) datasets. This would lead to an unfair comparison of gene expression specificity between the cell line diseases and single-cell types. As suggested by R1 (comment 4), we have removed the tau score paragraphs from the revised manuscript.

5. The section presenting the analysis of cancer-related pathways and cytokine activity in human cell lines is somewhat speculative. Here again, the authors should provide additional experimental evidence in support of their claim that specific cancer-related pathways and cytokines are upregulated in certain cancer cell lines. Alternatively, authors should consider presenting validation by analyzing data from an independent cohort (cell-line RNA-seq data that is not part of HPA and CCLE). Also, just mere up-regulation of certain pathways and cytokines does not fully justify the biological context. To enable the readers to interpret the results in a biologically meaningful manner, it is necessary to analyze these pathways and cytokines in TCGA tissue samples as well and try to assess the concordance of these pathways and cytokines in the context of the tissue microenvironment. The discordant pathways and cytokines between cell-lines and corresponding TCGA samples can either be cell-line specific (therefore not activated in TCGA samples) or related to the microenvironment (absent in the cell lines).

This is an interesting question. First, to validate the cancer-related pathways and cytokines activity, we calculated pathway/cytokine activities on an independent cancer cell line RNA-seq dataset from Genentech, Inc.⁴ ($n = 610$, with 461 overlapping with the cell lines analyzed in the manuscript) to validate the reliability of the results presented in the manuscript. Based on the mean square error (MSE) comparing pathway/cytokine activities between each pair of the common cell lines from HPA+CCLE and Genentech, results showed that the identical cell lines from different datasets have very similar pathway/cytokine profiles. The results have been added on page 9, line 314-319.

Second, the pathway/cytokine activities were also estimated for TCGA cohorts. Based on the PROGENy/CytoSig activities, we again calculated the similarity between cancer cell lines and the corresponding TCGA cohorts based on the mean square error (MSE), and prioritized top-5 (including tied) cell lines based on the combined ranking of PROGENy and CytoSig activities. We found that 31 of the total 112 pathway- and cytokine-based selected cell lines can be found in the transcriptomics-based selected cell lines (Figure 4) (hypergeometric testing $p = 4.14E-07$), suggesting a significant overlap of the prioritized cell lines between pathway/cytokine analysis and transcriptomics analysis. Of note, the PROGENy and CytoSig were designed to investigate the relative activities of the analyzed pathways and cytokines in the samples within one dataset^{5,6}. To ensure a fair and unbiased comparison, the HPA+CCLE, Genentech cell line datasets, and TCGA datasets were processed and analyzed separately, with different standard levels to represent the relative activities of the cell lines/tumors in each dataset (this is suggested by the authors in their original publications^{5,6}). This may exaggerate the differences in the pathway/cytokine activities between these three datasets. Even though, the significant overlap of the selected cell lines demonstrated the usage of the two computational tools and highlighted the value of presenting pathway and cytokine activities for the analyzed cell lines. This analysis has been added on page 9, line 319-325.

While these results are meaningful, we did not further dig into the pathway-/cytokine-based cell line selection, but used it as a validation for the two computational tools. The main reason is that the 14 pathways and 43 cytokines, despite being informative, may not cover the full transcriptomics information. For example, in the revised Figure 5 we demonstrated that some fundamental metabolic processes in liver

tissue were underexpressed in lowly-ranked liver cancer cell lines. This key information for cell line prioritization may not be well-reflected by pathway and cytokine analysis.

6. I agree that the author attempted to answer the question as stated in the title. However, the methods applied by the authors only involve transcriptomics analysis and lack any phenotypic and histologic interrogations. Therefore, it is necessary to state the specific method (eg: Transcriptomics) explicitly in the title that was used in the study to address the question. I would recommend changing the title to a more conventional one highlighting the transcriptomics analysis.

We thank the reviewer for pointing out this issue, which has also been proposed by reviewer #1 (comment 1) and #3 (comment 1). Based on the suggestions, we changed the title to "Systematic transcriptional analysis of human cell lines for gene expression landscape and tumor representation" to highlight that the study has been performed based on transcriptomics analysis.

Minor points:

1. Line: 145-148: How the enrichment was done?

The enrichment of gene expression in cell line dataset was calculated based on Table 1. For clarity's sake, we have added a note on page 5, line 153 to page 6, line 162.

"Subsequently, protein-coding genes were categorized into five different groups according to the expression specificity in CLDs (Table 1), including (i) CLD-enriched genes with at least fourfold higher expression levels (based on nTPM values) in one CLD as compared with any other analyzed CLD; (ii) group-enriched genes with enriched expression in a few CLDs (2 to 10); (iii) CLD-enhanced genes with only moderately elevated expression; (iv) low CLD specificity genes showing elevated expression in at least one of the analyzed CLDs; and (v) not detected genes at the CLD level. Similarly, we analyzed the TCGA transcriptomics dataset including 6,082 primary tumors from 26 cohorts with high tumor purity scores (> 0.7), and applied gene expression specificity classification to the TCGA dataset at the cohort level, which enabled the comparison of the gene expression specificity between CLDs and the TCGA cohorts."

2. Line: 118-119: What is number of common genes expressed in all CCLE and HPA cell lines?

We checked the number of common genes expressed in all CCLE and HPA cell lines and found that 5,209 of the total 5,366 genes expressed in all CCLE cell lines were also expressed in all HPA cell lines. We have added this information on page 5, line 129-131.

"In addition, we found that 5,366 (with 5,209 overlapping with those 6,799 genes HPA; Fig. 2a) genes are expressed in all 1,019 CCLE cell lines (nTPM > 1), suggesting these genes are essential in the cell line models."

3. Line: 124: Include the reference of GSOA.

We have added a reference of GSOA.

4. 143-144: What are the resources? For clarity, it is better to explicitly state the resources.

We have added a note to clarify the data we previously analyzed and published. Changes can be found on page 6, line 186-189.

"The tissue-enriched and single-cell type-enriched genes were derived from 56 tissues and 51 cell types from 13 human tissues (both are non-disease), respectively, with similar gene expression specificity strategies applied."

5. Line: 347-348: Is any reference available showing that PC-3 is indeed not suitable for androgen suppression therapy?

Indeed, many studies (PMID: 12858358; PMID: 32531951) used PC-3 with the re-expression of androgen receptor (PC3-AR) to study the mechanisms of androgen deprivation therapy. We have added the two references showing the differences between PC3 and PC3-AR on page 10, line 360.

Reviewer #3 - Computational tools, genomics - (Remarks to the Author):

The authors investigate the concordance between a large panel of cancer cell lines and tumours from a transcriptional point-of-view. This is a long-posed question that has been investigated in previous studies in the context of individual cancer types or in a pan cancer manner. To this purpose, the authors assess the transcriptional specificity of measured genes in cell lines and primary tumours as well as single-cell data from healthy tissues. In this way the authors report a reasonable enrichment of cell line cancer-specific genes which is concordant to what observed in TCGA primary tumour samples. They also performed an interesting analysis comparing metastatic cell lines expression with primary tumours of the same tissue or the same site. Additionally, they merged two gene expression based criteria to find the top 5 cell lines representing a cancer type and finally investigated pathway and cytokine activity scores at the cell line level to inform on phenotypic features that could be used in in-vitro drug screenings.

Overall, this is an interesting work tackling an important problem. The manuscript is well-structured and provides a valuable tool for the selection of cell-line most transcriptionally similar to primary tumours.

However, this topic has already been extensively studied and several tools are now available pursuing aims similar to those of motivating this study. The authors fail to highlight the innovative aspects of the proposed method/analysis. Particularly, a systematic comparison with previous results from tools for the selection of good-quality cell lines should be compulsorily added important.

In addition, the following points should be addressed.

1. The title is a question that is never answered throughout the text. The authors should draw some conclusive remarks in the form of answers to the title. In addition, it reads odd, I would suggest to change it in: "Are cell lines good models of primary tumours?" an alternative would be to opt for a more descriptive title that is reflective of the performed analyses and results. Something along the lines "Systematic transcriptional analysis of cell lines ... etc"

We thank the reviewer for the suggestion on retitling the manuscript, which has also been proposed by reviewer 1# (comment 1) and #2 (comment 6). Based on the suggestions, we changed the title to "Systematic transcriptional analysis of human cell lines for gene expression landscape and tumor representation" to better reflect the performed analyses and results.

2. The authors claim that "a comprehensive comparison of gene expression between cell lines, human tissues, tumours and cells is still lacking". They should mention that such comprehensive comparison has already been performed, at least between cell lines and tumours in PMID: 33397959 and put more emphasis on the difference between this work and their analysis, which is original because is extended to normal tissues and individual cells. This would also help to highlight the novelty of their work generally.

Following the suggestion, we have rephrased the introduction part. Changes can be found on page 3, line 53-58.

"Despite a few studies that have attempted to map cancer cell lines to the corresponding diseases based on the molecular signatures, which facilitates the selection of appropriate cell lines for cancer research, these studies did not further extend their comparison of cell line gene expression to normal tissues and individual cell types – a critical step which will help us to understand the characteristics and representative of cell lines to their origin."

3. In PMID: 33397959, the authors report a 'central cluster' cell lines presenting a more mesenchymal and undifferentiated transcriptional state. I can't see any reference to this finding. It is important that the authors mention this and characterise in dept these cell lines.

We thank the reviewer for the suggestions for improving the manuscript. We have carefully examined these cell lines and made a comparison in our showcase for liver cancer cell lines. We found that only one of the first 12 cell lines (i.e., SNU-886, ranked 12th) in our ranking list were reported as "undifferentiated", whereas 9 of the last 12 cell lines were "undifferentiated". This could explain why some cell lines were lowly ranked (most of them are "undifferentiated"). We have referred to this study and added the comparison results on page 8, line 273-278.

“Indeed, a previous study reported that some cell lines presented an undifferentiated state and may be derived from undifferentiated tumor. In our ranking list for the 24 liver cancer cell lines, only the SNU-886 (ranked 12th) was reported as “undifferentiated” in the first 12 cell lines, but 9 of the last 12 cell lines were reported as “undifferentiated”. The imbalance between highly- and lowly-ranked cell lines regarding their differentiated state possibly explains why some cell lines are lowly prioritized.”

4. Other important previous studies similar to what presented in this manuscript should be cited and briefly described, again highlighting differences and commonalities with what the authors present here: particularly Sinha et al (2021) (TumorComparer), Zhang & Kschischo (2021) (MFmap) and Salvadores et al (2020) (HyperTracker)

These studies have been cited and briefly discussed in the discussion section (page 18, line 2-6). Considering the word limit, and the methods used in these different studies are very distinct, we could not compare our methods with all the studies mentioned above, but in the revised manuscript we do compare our method to a previous study which is also based on correlation analysis (Yu et al. TCGA-110-CL³) and demonstrated the advantage and improvement of our study (please see reply to comment 17). These results have been added to the manuscript (page 8, line 250-255; page 11, line 379-382).

“To validate the selected candidates, we compared the results with a previous study using 5,000 most variable genes for correlation-based cell line selection (TCGA-110-CL; n = 100 selected cell lines for the same cohorts analyzed in this study). We found that 65 of our selections were also included in the TCGA-110-CL panel, with a p-value of 1.55E-44 based on hypergeometric testing (Supplementary Fig. 6a), underpinning the validity of the selected cell lines by our approach.”

“Especially the adoption of the GSEA, which showed discriminative in estimating fundamental tumor biological processes in cell lines (Figure 5c), has not yet been investigated in the previous correlation-based study.”

5. The lack of batch effects between the HPA and CCLE transcriptional datasets comes out clearly from supplementary figure 2, although it should be quantified. Particularly, the authors claim that cell lines in common across the two datasets lie ‘tightly’ close to each other in the integrated UMAP space. This qualitative claim is not acceptable and the authors should prove a more robust estimation such as for example percentage of cell lines in one dataset that have as 1st, 2nd, 3rd, ... closest neighbor their counterpart in the other dataset. The dendrogram in Supplementary figure 3 is not enough. In addition, in that figure, most of the cell lines shared across the two datasets have their two labels superimposed, creating an annoying anaglyph-3D-like effect. The authors should work on an alternative visualisation.

We thank the reviewer for his/her rigorous comment. Following the suggestion, we calculated the Euclidean distance of the common cell lines between CCLE and HPA based on the PCA dimensions (nPCs = 398). We found that for the 33 common cell lines in CCLE, 32 have the first neighbor in the HPA database. Only cell line U-251MG was the 4th neighbor in the HPA database. This demonstrated the closeness of each pair of the common cell line between CCLE and HPA. We have added the result on page 4, line 90-93.

“To further confirm the closeness of these common cell lines, based on the 398 PCs, we calculated the Euclidean distance from CCLE to HPA cell lines, and found that for the 33 common cell lines, 32 have their counterpart as the first neighbor in HPA.”

We have also revised Supplementary Figure 3a to improve the visualization.

6. Similar considerations apply to the qualitative observations that the authors made on cell lines from certain cancer type being overall more similar/dissimilar to the others, based on figure 1c. These overall similarities should be quantified, for example computing the ratio between inter/intra-cluster variance for each cancer type.

To address this concern, we calculated the correlation based on all available protein-coding genes between the cell lines from the same cancer type (intra-cluster variance) compared to the cell lines from different cancer types (inter-cluster variance). Results show that cell lines from the same cancer type have

significantly higher correlations than the cell lines from different cancer types. We have updated the results in Supplementary Figure 3b (page 5, line 112-113).

“Indeed, this was also demonstrated by the significantly higher correlations between the cell lines from the same disease than the ones from different diseases (Supplementary Fig. 3b).”

7. Again, the authors claim: “more than half of the cell lines were grouped by cancer types, suggesting that cell line preserve the cancer phenotype at the transcriptional level”. This claim is not supported by any quantitative measurement applied to the clusters in figure 1c, their ability to correctly classify a single cancer type, nor statistical enrichment of cancer types across clusters, ROC indicators. Deriving such biological conclusions just by eyeballing a uMAP plot is not acceptable.

We agree that this claim based on just eyeballing is vague. To investigate the ability to correctly classify the cancer types based on cell line gene expression, we applied machine learning prediction models logistic regression (LR) and random forests (RF) with stratified 5-fold cross-validation with 100 repeats on the cell line gene expression data. To realize the cross-validation setting, only a cancer type harboring at least five cell lines was selected for classification analysis, resulting in 26 cancer types selected for testing the classification ability of the cell line expression data. LR and RF achieved 74% and 68% accurate rates, suggesting that more than half of the cell lines can be correctly classified by cancer type. We have updated the results on page 5, line 113-116.

“In addition, based on the top 5000 most variable genes, 74% and 68% of the cell lines can be correctly classified (5-fold cross-validation) for their cancer types by logistic regression and random forests, respectively.”

8. The authors reference Table 1 to explain how they have explored gene expression distributions and specificity across cancer type. Unfortunately this table is not clear at all. They should briefly but clearly describe their analysis in the main text.

We found that the descriptions about the gene expression specificity were not clear enough, therefore we have revised Table 1 and added descriptions in the main text (page 5, line 153 to page 6, line 162) to improve the readability.

“Subsequently, protein-coding genes were categorized into five different groups according to the expression specificity in CLDs (Table 1), including (i) CLD-enriched genes with at least fourfold higher expression levels (based on nTPM values) in one CLD as compared with any other analyzed CLD; (ii) group-enriched genes with enriched expression in a few CLDs (2 to 10); (iii) CLD-enhanced genes with only moderately elevated expression; (iv) low CLD specificity genes showing elevated expression in at least one of the analyzed CLDs; and (v) not detected genes at the CLD level. Similarly, we analyzed the TCGA transcriptomics dataset including 6,082 primary tumors from 26 cohorts with high tumor purity scores (> 0.7), and applied gene expression specificity to the TCGA dataset at the cohort level, which enabled the comparison of the gene expression specificity between CLDs and the TCGA cohorts.”

9. What is the point of highlighting transcriptional differences across CCLE and HPA? (figure 2a)

Here we would like to demonstrate that compared with the HPA cell lines (n = 69), the 1,019 CCLE cell lines showed a lower number of non-detectable genes and genes specifically expressed in a single cell line (both because of the increased number of the cell lines) and could reflect a broader coverage of gene expression, thus by adding the CCLE cell lines into the HPA database, the updated HPA cell line section has been significantly expanded and could reflect a more comprehensive gene expression landscape. To let readers better understand our aim, we have added descriptions in the main text (page 5, line 126-129).

“Hence, by incorporating the CCLE dataset into the HPA database to significantly increase the total number of cell lines, we found that the number of cell line-specific or not detected genes became smaller, suggesting that the integration of the two cell line datasets may reflect a broader coverage of gene expression.”

10. The authors draw conclusions from comparing housekeeping genes and ‘essential’ genes but

referencing not very recent works based on CRISPR-screening very few cell lines. Much more comprehensive and more recent CRISPR/shRNA screens have been performed on thousands of cancer cell lines (potentially covering most of the CCLC cell lines) and should be used for these comparisons (for example, the works/datasets presented in: PMID: 28753430, PMID: 30971826, PMID: 33712601).

We thank the reviewer for pointing out this issue. Following the suggestion, we have redone the analysis using new data from DepMap CRISPR gene screening results¹⁷. For this, we downloaded the file `depmap_22Q2_crispr_common_essentials.csv` (based on 22Q2 version) from the webpage <https://depmap.org/portal/download/all/> and used the 1,912 essential genes for intersect analysis. Overall, the results are highly consistent with previous results, and the main conclusion remains unchanged. We have updated the manuscript based on the new DepMap CRISPR data (Fig. 2b-c; line 134-140 on page 5).

“In this context, we interrogated the 5,366 genes expressed in all 1,019 CCLC cell lines with the previously reported 1,912 essential genes based on CRISPR gene screening results (DepMap 22Q2, CRISPR_common_essentials.csv from <https://depmap.org/portal/download/all/>), and found that 1,614 genes were overlapped between two different datasets. Based on gene set overrepresentation analysis (GSOA), these genes were strongly associated with DNA replication and nuclear division, suggesting these genes are indispensable for cell cycle progression and cell proliferation (Fig. 2c, Supplementary Table 3).”

11. The choice of ‘cell line cancer type’ to refer to transcriptional profiles resulting from cell line aggregation on a cancer type level is quite unfortunate and confusing. The authors should opt of an alternative, even an acronym would be better.

Following the suggestion, we have termed cell line cancer type as CLD (cell line disease). See page 5, line 152-153, and others.

“Hereafter, we termed CLD (cell line disease) to represent cell lines grouped by the same cancer type.”

12. The analysis summarised in figure 2d is well conceived but, again, results are summarised in a qualitative rather than quantitative way and this should be amended. What is the gene grouping similarity across categories between Cell line cancers and TCGA cancers? There are suitable metrics to assess this.

We thank the reviewer for pointing out this issue. To better reflect the similarity of gene expression specificity between cell line diseases and TCGA cohorts, we calculated the percentage of the genes being classified as the same specificity category relative to the total analyzed genes and found that 66.47% had the same categorization. Results have been added on page 6, line 169-173.

“The number of genes differently distributed in CLDs is similar to what was observed in the 26 TCGA cohorts (Fig. 2d), with 13,331 (66.47%) genes being classified as exactly the same category between CLDs and TCGA cohorts, suggesting a high similarity of gene expression distribution and specificity between cell line and TCGA datasets.”

13. The authors find that high specific genes identified in cell line cancer types but not in tumours are enriched for immune-related biological processes. This is very puzzling, as cell line are lacking immune components. The inverse associations would have been more plausible. The authors should comment and justify this.

We agree with the reviewer that the immune-related biological processes are expected to be enriched in TCGA samples but not cell lines. The main reason is that the cell line dataset has blood cancer cell lines (myeloma, lymphoma, leukemia) whereas in the original manuscript, we did not include blood cancer in TCGA analysis. To overcome this pitfall, we have reanalyzed the TCGA data, increasing the number of cohorts analyzed from 21 to 26 to align the TCGA cohorts with cell line cancer types at the best efforts. After expanding the number of analyzed TCGA cohorts (especially with blood cancer cohort LAML included), this imbalance issue has been solved, thus the results in the original manuscript (the original Figure S4a-c and the corresponding text) were removed.

14. cell lines are grouped by their primary disease, but it was shown in previous studies (e.g. Salvadores et al (2020)) that some lines are mislabelled or transdifferentiated. I suggest looking at the mentioned publication and add a comment on it.

Following the suggestion, we have added a note in the discussion (page 12, line 417-419) to indicate the historical issue of the cell line annotation. For the cell lines analyzed in this study, we provided RRID to link them to the information page in the Cellosaurus database (<https://www.cellosaurus.org/>). This will help users to further examine the detailed information when selecting appropriate cell line models for their research.

“Furthermore, we provided RRID for the cell lines in the HPA database, linking them to the information page of the Cellosaurus database. This will help users to bypass the selection of mislabeled or transdifferentiated cell lines due to historical reasons.”

15. in the comparison with TCGA, it would make more sense to use the common set of cancer type. Indeed, the different sets of high-specific genes arise from TCGA lack of blood cancer cohorts (line 176).

We agree. Therefore in the current manuscript, we expanded the number of TCGA cohorts from 21 to 26, with CHOL (bile duct cancer/cholangiocarcinoma), ESCA (esophageal cancer), LGG (brain cancer), and SARC (sarcoma), and blood cancer cohort LAML included, to align the TCGA cohorts with cell line cancer types at the best effort. Based on this, the reliability of the results presented in Figure 2d and others has been significantly improved.

16. From their analysis the authors conclude that gene specificity is similar. I understand it is relevant when looking at the same cancer type, but I do not get why it is useful when across data-set they do not even use the same set of tissue type.

The gene expression specificity landscape in human tissues was obtained from the tissue section of the Human Protein Atlas (HPA) database, which contains mRNA expression from more than 50 normal tissue types across the whole body¹⁸. The aim of the HPA project is to investigate gene expression and protein abundance distributed in the whole body, which allows a global comparison between cell line gene expression specificity and tissue¹⁸ and single-cell type¹⁶. Since the comparison is based on the whole body, most of the tissues/cancers/single-cell types were overlapped. To avoid confusion to the audience, we did not perform redundant gene expression specificity analyses based on a slightly different set of tissues or single-cell types. Despite not being perfectly matched, our results showed that gene specificity in cell lines was similar to the corresponding tissues, TCGA cancers, and single-cell types (Figure 3a and Supplementary Figure 4).

17. correlation methods to match cell lines and tumours have been widely used (many reviewed in PMID: 35822563). Since the raking problem from transcriptome was also previously addressed, it would be interesting to see a comparison with previous studies (e.g. Yu et al (2019) ranking results).

We thank the reviewer for this constructive suggestion. In the revised manuscript, we compared our results with competing methods from independent studies, i.e., the TCGA-110-CL panel provided by Yu et al.³

After aligning the cell line diseases and TCGA cohorts with Yu et al. study, we found that 65 of the total 114 selected cell lines in this study are presented in the TCGA-110-CL panel (n = 100) (hypergeometric testing $p = 1.55E-44$). This significant overlap supports the validity of the selected cell lines in our study. Of note, Yu et al. study only relies on the correlation based on the 5000 most variable genes, while our analysis is transcriptome-wise based using more than 20,000 protein-coding genes – a significant improvement compared to the previous study. The other advantage of our study is the use of GSEA to estimate the enrichment of the TCGA disease signatures, which have not been considered in the previous study. The validity of GSEA has been demonstrated in the new Figure 5, showing that some liver cancer cell lines have weaker disease feature compared to others. Taken together, our 2D (correlation-based and GSEA-based) cell line prioritization method outperforms the previous correlation-based studies.

The results discussed above have been added to the manuscript (page 8, line 250-255; page 11, line 379-382).

“To validate the selected candidates, we compared the results with a previous study using 5,000 most variable genes for correlation-based cell line selection (TCGA-110-CL; n = 100 selected cell lines for the same cohorts analyzed in this study). We found that 65 of our selections were also included in the TCGA-110-CL panel, with a p-value of 1.55E-44 based on hypergeometric testing (Supplementary Fig. 6a), underpinning the validity of the selected cell lines by our approach.”

“Especially the adoption of the GSEA, which showed discriminative in estimating fundamental tumor biological processes in cell lines (Figure 5c), has not yet been investigated in the previous correlation-based study.”

18. The pathway and cytokine based analysis has been very nicely performed and presented in a set of effective visualisations. It would be interesting to see how progeny and cytosign features are actually distributed in TCGA and compared cancer specific scores with cell lines (similar to what was done in paragraph 4 for spearman correlation), would be the top 5 cell lines also highly concordant from this new features?

This is a very interesting question. Following the suggestions, the pathway/cytokine activities were also estimated for TCGA cohorts. Based on the PROGENY/CytoSig activities, we again calculated the similarity between cancer cell lines and the corresponding TCGA cohorts based on the mean square error (MSE), and prioritized top-5 (including tied) cell lines based on the combined ranking of PROGENY and CytoSig activities. We found that 31 of the total 112 pathway- and cytokine-based selected cell lines can be found in the transcriptomics-based selected cell lines (Figure 4) (hypergeometric testing $p = 4.14E-07$), suggesting a significant overlap of the prioritized cell lines between pathway/cytokine analysis and transcriptomics analysis. Of note, the PROGENY and CytoSig were designed to investigate the relative activities of the analyzed pathways and cytokines in the samples within one dataset^{5,6}. To ensure a fair and unbiased comparison, the HPA+CCLE, Genentech cell line datasets, and TCGA datasets were processed and analyzed separately, with different standard levels to represent the relative activities of the cell lines/tumors in each dataset (this is suggested by the authors in their original publications^{5,6}). This may exaggerate the differences in the pathway/cytokine activities between these three datasets. Even though, the significant overlap of the selected cell lines demonstrated the usage of the two computational tools and highlighted the value of presenting pathway and cytokine activities for the analyzed cell lines. This analysis has been added on page 9, line 319-325.

While these results are meaningful, we did not further dig into the pathway-/cytokine-based cell line selection, but used it as a validation for the two computational tools. The main reason is that the 14 pathways and 43 cytokines, despite being informative, may not cover the full transcriptomics information. For example, in the revised Figure 5 we demonstrated that some fundamental metabolic processes in liver tissue were underexpressed in lowly-ranked liver cancer cell lines. This key information for cell line prioritization may not be well-reflected by pathway and cytokine analysis.

19. Lines 355-357: I think this is overstated, previous studies have proposed a subset of cell lines as “good” models using gene expression data. For example, Peng et al (2021) (CancerCellNet), Yu et al (2019) (CompHealth) , Salvadores et al (2020). I would stress instead the advantages of the proposed study compared to the previous one

We acknowledge that the relevant sentence is overstated. Accordingly, we revised the sentence as below to highlight the global comparison between cell lines to the tissues, TCGA cancers and single-cell types across the whole body, which is a significant step forward compared to the previous studies that only focused on the comparison between cell lines and cancers (page 11, line 366-369).

“While these studies have substantially deepened our understanding of the cell line representative of human cancers, they mostly lack answering a fundamental question, that is, how well is the representative of cell lines to the corresponding tissues, tumors, and cell types.”

In addition, the advantage of our approaches for the cell line section has been discussed on page 11, line 16-19. A comparison between our results with a competing method based on correlation analysis (Yu et al. TCGA-110-CL³) has also been provided (page 8, line 250-255).

“To validate the selected candidates, we compared the results with a previous study using 5,000 most variable genes for correlation-based cell line selection (TCGA-110-CL; n = 100 selected cell lines for the same cohorts analyzed in this study). We found that 65 of our selections were also included in the TCGA-110-CL panel, with a p-value of 1.55E-44 based on hypergeometric testing, underpinning the validity of the selected cell lines by our approach.”

Other minor points:

20. While referencing previous works aiming at identifying new therapeutic targets based on functional/chemo-genomics screens of large panels of immortalised human cell lines, the following works could be cited: PMID: 28753430, PMID: 30971826

We have replaced the old referenced (published in 2015) with these two studies on page 5, line 134.

21. Large cell line-based Multi-omic datasets and their use to identify therapeutic biomarkers could also be mentioned. On this regard, the following works might be cited: PMID: 23180760, PMID: 27397505, PMID: 22460902

We have discussed the potential of integrating these resources as well as cell line proteomics and metabolomics data to make multi-omics study for future directions (page 12, line 428-431).

“Future works would focus on multi-omics analysis of cell lines using genomics and recently released proteomics, metabolomics and drug response datasets to comprehensively evaluate and expand the main findings in this study.”

22. Fig. 2e is never references in the main text

We have referred to this figure panel on page 6, line 175.

References

- 1 Nwosu, Z. C. *et al.* Liver cancer cell lines distinctly mimic the metabolic gene expression pattern of the corresponding human tumours. *Journal of Experimental & Clinical Cancer Research* **37**, 211, doi:10.1186/s13046-018-0872-6 (2018).
- 2 Yoshihara, K. *et al.* Inferring tumour purity and stromal and immune cell admixture from expression data. *Nature Communications* **4**, 2612, doi:10.1038/ncomms3612 (2013).
- 3 Yu, K. *et al.* Comprehensive transcriptomic analysis of cell lines as models of primary tumors across 22 tumor types. *Nature Communications* **10**, 3574, doi:10.1038/s41467-019-11415-2 (2019).
- 4 Klijn, C. *et al.* A comprehensive transcriptional portrait of human cancer cell lines. *Nature Biotechnology* **33**, 306-312, doi:10.1038/nbt.3080 (2015).
- 5 Schubert, M. *et al.* Perturbation-response genes reveal signaling footprints in cancer gene expression. *Nature Communications* **9**, 20, doi:10.1038/s41467-017-02391-6 (2018).
- 6 Jiang, P. *et al.* Systematic investigation of cytokine signaling activity at the tissue and single-cell levels. *Nature Methods* **18**, 1181-1191, doi:10.1038/s41592-021-01274-5 (2021).

- 7 Robinson, M. D. & Oshlack, A. A scaling normalization method for differential expression analysis of RNA-seq data. *Genome Biology* **11**, R25, doi:10.1186/gb-2010-11-3-r25 (2010).
- 8 Konieczkowski, D. J. *et al.* A Melanoma Cell State Distinction Influences Sensitivity to MAPK Pathway Inhibitors. *Cancer Discovery* **4**, 816-827, doi:10.1158/2159-8290.Cd-13-0424 (2014).
- 9 Singh, A. *et al.* A Gene Expression Signature Associated with “K-Ras Addiction” Reveals Regulators of EMT and Tumor Cell Survival. *Cancer Cell* **15**, 489-500, doi:<https://doi.org/10.1016/j.ccr.2009.03.022> (2009).
- 10 Kim, J. W. *et al.* Decomposing Oncogenic Transcriptional Signatures to Generate Maps of Divergent Cellular States. *Cell Systems* **5**, 105-118.e109, doi:<https://doi.org/10.1016/j.cels.2017.08.002> (2017).
- 11 Miller, L. D. *et al.* An expression signature for p53 status in human breast cancer predicts mutation status, transcriptional effects, and patient survival. *Proceedings of the National Academy of Sciences* **102**, 13550-13555, doi:doi:10.1073/pnas.0506230102 (2005).
- 12 Gonçalves, E. *et al.* Pan-cancer proteomic map of 949 human cell lines. *Cancer Cell*, doi:<https://doi.org/10.1016/j.ccell.2022.06.010> (2022).
- 13 Li, H. *et al.* The landscape of cancer cell line metabolism. *Nature Medicine* **25**, 850-860, doi:10.1038/s41591-019-0404-8 (2019).
- 14 Salvadores, M., Fuster-Tormo, F. & Supek, F. Matching cell lines with cancer type and subtype of origin via mutational, epigenomic, and transcriptomic patterns. *Science Advances* **6**, eaba1862, doi:doi:10.1126/sciadv.aba1862 (2020).
- 15 Sinha, R., Luna, A., Schultz, N. & Sander, C. A pan-cancer survey of cell line tumor similarity by feature-weighted molecular profiles. *Cell Reports Methods* **1**, 100039, doi:<https://doi.org/10.1016/j.crmeth.2021.100039> (2021).
- 16 Karlsson, M. *et al.* A single-cell type transcriptomics map of human tissues. *Science Advances* **7**, eabh2169, doi:doi:10.1126/sciadv.abh2169 (2021).
- 17 Tsherniak, A. *et al.* Defining a Cancer Dependency Map. *Cell* **170**, 564-576.e516, doi:<https://doi.org/10.1016/j.cell.2017.06.010> (2017).
- 18 Uhlén, M. *et al.* Tissue-based map of the human proteome. *Science* **347**, 1260419, doi:doi:10.1126/science.1260419 (2015).

Reviewers' Comments:

Reviewer #1:

Remarks to the Author:

The authors have adequately addressed many of the points raised. However, several points remain open and the novelty of the study is still not convincingly demonstrated. In line with that several of the statements made remain overstate.

Specific points that remain to be addressed are listed below:

In point 6 I criticized the significance of the spearman correlation, as more than 5000 proteins are observed in all cell lines and the relative differences in the scores are small. The response of the authors was : "In the revised manuscript, with the inclusion of more TCGA cohorts (especially the blood cancer cohort LAML), correlations between cell line diseases and TCGA cohorts are more distinguishable, ranging from 0.655 to 0.852. In addition, we changed the color palette for the heatmap to provide a more eye-comfortable gradient thus high correlations are further highlighted."

While the authors indeed improved the visualization of the data and expanded their analysis, the main issue remains that the significance of the analysis is not clear. As stated the correlations of the samples range from 0.65 and 0.85, but it remains unclear how significant these relative differences are to demonstrate the point the authors try to make that the cell lines are closer correlated to the tissue they were derived from. In other words, the significance of the different spearman correlation coefficient should be tested. For example: based on the color scale the correlation of lung cancer cell lines with liver cancer (LIHC) is around 0.73. The correlation with LUSC and LUAD (lung cancer) is about 0.8 and it is unclear whether this is significant.

Furthermore to highlight the novelty of their study the authors state "Compared with the previous study (5000 genes), a transcriptome-wise analysis based on more than 20,000 protein-coding genes is the advantage of this study."

However, while it is evident that including 15000 genes more in a study could be more informative, it is not clear what the specific novel findings of the presented study are. Thus, it is important to precisely explain how the findings differ from or improve previous knowledge and to specify the relevance for future research.

In point 9 we explained that the analysis of cytokine expression and pathway activation primarily confirmed that the employed software packages worked but did not demonstrate to which extent novel findings were obtained. The authors respond: "Here we provided such information for more than 1,000 cell lines to facilitate researchers' selection of the appropriate cell lines for basic research, clinical and translational purposes. The meaning of providing such information was demonstrated in Figure 7 and discussed in the third paragraph of the discussion section. To further highlight the novelty of the pathway/cytokine analysis, in the revised manuscript, we performed extra analyses to make this part more technically robust and scientifically meaningful...While these results are meaningful, we did not further dig into the pathway-/cytokine-based cell line selection, but used it as a validation for the two computational tools. The main reason is that the 14 pathways and 43 cytokines, despite being informative, may not cover the full transcriptomics information. For example, in the revised Figure 5 we demonstrated that some fundamental metabolic processes in liver tissue were underexpressed in lowly-ranked liver cancer cell lines. This key information for cell line prioritization may not be well-reflected by pathway and cytokine analysis."

To confirm the novelty of their findings, the authors generated based on available data and utilizing established computational tools a collection of profiles for 1000 cell lines. Although these profiles might be useful for others, the novelty of the study is still not convincing as the data and the analysis tools were already available. Importantly, it is claimed that information on "underexpressed" pathways in the cell lines is displayed in Figure 5, but this statement is not supported by the data and thus should either be toned down or better explained.

Reviewer #2:

Remarks to the Author:

In the revised version of their manuscript entitled "Systematic transcriptional analysis of human cell lines for gene expression landscape and tumor representation", as well as in the point-by-point responses, the authors have clearly addressed my remarks to the earlier version of the

manuscript. As such, I support the publication of the present version of the manuscript.

Reviewer #3:

Remarks to the Author:

The authors have demonstrated commendable dedication in addressing both my feedback and that of other reviewers. Consequently, the manuscript has undergone significant enhancements, making the results and analysis considerably more robust. I am pleased with the overall improvements achieved in this new version.

The following review might be cited in the introduction while discussing computational methods for estimating quality and clinical relevance of cancer cell lines: PMID: 35822563

****REVIEWER REPORTS****

Reviewer #1

The authors have adequately addressed many of the points raised. However, several points remain open and the novelty of the study is still not convincingly demonstrated. In line with that several of the statements made remain overstate.

We thank the reviewer for his/her rigorous thought and efforts in improving the scientific quality of this study. We would like to take this opportunity, together with the reviewer, to improve the study substantially and make it more robust. Following the suggestions below, we have carefully revised the manuscript.

Specific points that remain to be addressed are listed below:

In point 6 I criticized the significance of the spearman correlation, as more than 5000 proteins are observed in all cell lines and the relative differences in the scores are small. The response of the authors was : "In the revised manuscript, with the inclusion of more TCGA cohorts (especially the blood cancer cohort LAML), correlations between cell line diseases and TCGA cohorts are more distinguishable, ranging from 0.655 to 0.852. In addition, we changed the color palette for the heatmap to provide a more eye-comfortable gradient thus high correlations are further highlighted."

While the authors indeed improved the visualization of the data and expanded their analysis, the main issue remains that the significance of the analysis is not clear. As stated the correlations of the samples range from 0.65 and 0.85, but it remains unclear how significant these relative differences are to demonstrate the point the authors try to make that the cell lines are closer correlated to the tissue they were derived from. In other words, the significance of the different spearman correlation coefficient should be tested. For example: based on the color scale the correlation of lung cancer cell lines with liver cancer (LIHC) is around 0.73. The correlation with LUSC and LUAD (lung cancer) is about 0.8 and it is unclear whether this is significant.

We thank the reviewer for this thoughtful suggestion and elaborated explanation. Following the suggestion, for each CLD, we use one-tailed one-sample Wilcoxon signed-rank test to investigate if its correlations to unmatched TCGA cohorts are significantly lower than its correlation to the matched TCGA cohort. Based on the CLD-TCGA disease matching table (see Supplementary Table 4), a total of 26 statistical tests were performed, with 23 of them being significant. Our results demonstrated that cell lines are generally correlated to the tumor they were derived from. We have updated Figure 3c and the corresponding figure legends. See below:

Changes in the figure legend can be found on page 20:

“For each CLD, we used one-tailed one-sample Wilcoxon signed-rank test to investigate if the correlations to its unmatched TCGA cohorts were significantly lower than the correlation to its matched TCGA cohort. Based on the information in Supplementary Table 4, 26 statistical tests were performed. * $P < 0.05$.”

Furthermore to highlight the novelty of their study the authors state “Compared with the previous study (5000 genes), a transcriptome-wide analysis based on more than 20,000 protein-coding genes is the advantage of this study.”

However, while it is evident that including 15000 genes more in a study could be more informative, it is not clear what the specific novel findings of the presented study are. Thus, it is important to precisely explain how the findings differ from or improve previous knowledge and to specify the relevance for future research.

Our consideration for using all protein-coding genes for the analysis is twofold. First, for all the resources presented in the Human Protein Atlas project, we always focus on all protein-coding genes and transcriptome-wide analysis. We believe this will provide more comprehensive information than using a subset of genes. Second, in the paper we referred to¹, while using 5,000 most variable genes for their correlation analysis in the main text, the authors compared their results with those using all protein-coding genes (please see Supplementary Figure 3B in https://static-content.springer.com/esm/art%3A10.1038%2Fs41467-019-11415-2/MediaObjects/41467_2019_11415_MOESM1_ESM.pdf). Even though the results based on 5000 genes and all genes are highly correlated ($R = 0.9$), the subtle difference presented in their analyses should not be ignored, which may subsequently influence the ranking list obtained from cell line prioritization analysis. To verify this, here we use liver cancer cell lines and the TCGA-LIHC as an example, comparing the correlation ranking lists based on all protein-coding genes with that based on the top 5,000 variable genes selected based on IQR in the TCGA-LIHC cohort (which is the same to Yu et al. paper¹).

From the table below we can see, for the total 24 cell lines analyzed, only 10 of them are ranked the same. This demonstrated that using only a subset of all protein-coding genes may not precisely capture correlations between cell lines and TCGA tumors.

RRID	Cell line name	Rank (all genes)	Rank (top 5000 genes)	Consistent
CVCL_0027	Hep-G2	1	1	Yes
CVCL_0336	Huh-7	2	4	
CVCL_0364	JHH-5	3	3	Yes
CVCL_2956	HuH-1	4	2	
CVCL_5102	SNU-878	5	5	Yes
CVCL_0326	Hep 3B2.1-7	6	7	
CVCL_2805	JHH-7	7	9	
CVCL_5089	SNU-761	8	6	
CVCL_3840	Li-7	9	11	
CVCL_0485	PLC/PRF/5	10	10	Yes
CVCL_4381	HuH-6	11	8	
CVCL_5103	SNU-886	12	12	Yes
CVCL_2786	JHH-2	13	13	Yes
CVCL_0497	SNU-475	14	15	
CVCL_2788	JHH-6	15	17	
CVCL_0250	SNU-387	16	19	
CVCL_2785	JHH-1	17	14	
CVCL_0454	SNU-449	18	18	Yes
CVCL_2787	JHH-4	19	16	
CVCL_0525	SK-HEP-1	20	21	
CVCL_0366	SNU-423	21	20	
CVCL_0090	SNU-182	22	22	Yes
CVCL_2947	HLF	23	23	Yes
CVCL_0077	SNU-398	24	24	Yes

Considering the word limit, and that the results are less supportive to the main conclusion in this paper, we therefore tend not to include these results in the manuscript. Instead, we add a note in the manuscript on page 8 to stress that the analyses were based on all protein-coding genes:

“Of note, our study relied on all protein-coding genes rather than a subset of genes for correlation analysis, which could provide a more comprehensive and precise evaluation of the similarity between cell lines and tumors.”

In point 9 we explained that the analysis of cytokine expression and pathway activation primarily confirmed that the employed software packages worked but did not demonstrate to which extent novel findings were obtained. The authors respond: “Here we provided such information for more than 1,000 cell lines to facilitate researchers' selection of the appropriate cell lines for basic research, clinical and translational purposes. The meaning of providing such information was demonstrated in Figure 7 and discussed in the third paragraph of the discussion section. To further highlight the novelty of the pathway/cytokine analysis, in the revised manuscript, we performed extra analyses to make this part more technically robust and scientifically meaningful...While these results are meaningful, we did not further dig into the pathway-/cytokine-based cell line selection, but used it as a validation for the two computational tools. The main reason is that the 14 pathways and 43 cytokines, despite being informative, may not cover the full transcriptomics information. For example, in the revised Figure 5 we demonstrated that some fundamental

metabolic processes in liver tissue were underexpressed in lowly-ranked liver cancer cell lines. This key information for cell line prioritization may not be well-reflected by pathway and cytokine analysis.”

To confirm the novelty of their findings, the authors generated based on available data and utilizing established computational tools a collection of profiles for 1000 cell lines. Although these profiles might be useful for others, the novelty of the study is still not convincing as the data and the analysis tools were already available. Importantly, it is claimed that information on “underexpressed” pathways in the cell lines is displayed in Figure 5, but this statement is not supported by the data and thus should either be toned down or better explained.

We thank the reviewer for this suggestion regarding the pathway and cytokine activity. Despite the fact that the HPA and CCLE cell line RNA-seq datasets, as well as the two computational tools, are publicly available, the novelty of the results comes from the application of these well-developed analytical tools on these high-quality, large-scale cell line RNA-seq datasets. For example, many large RNA-seq datasets, including the TCGA, have been published these years, followed by the application of the developed analytical tools on these datasets studies to discover novel findings. For the results presented in this study, to our knowledge, there is currently no study that reported such comprehensive information about pathway and cytokine activity in more than 1,000 cell lines.

In this study, we generated novel findings by applying the developed analytical tools to the existing dataset, and we presented some biological findings that are well-known in some previous studies as positive controls (for example, significant/non-significant elevated androgen receptor activity corresponding to the AR-positive/-negative cell lines, see Figure 7c). Overall, our analysis provided a referential phenotype characterization of these 1,055 cell lines that cannot be easily obtained from *in vitro* experiments, and most of them have not been reported before, which we believe is the novelty of this study.

Lastly, following the last suggestion, we have toned down the sentences on page 8 for Figure 5:

“Meanwhile, the TCGA-LIHC signature was also found **relatively** lowly-expressed in the SNU-398 cell line, suggesting that the hallmark of TCGA-LIHC might be **weaker** in this cell line.”

Reviewer #2

In the revised version of their manuscript entitled "Systematic transcriptional analysis of human cell lines for gene expression landscape and tumor representation", as well as in the point-by-point responses, the authors have clearly addressed my remarks to the earlier version of the manuscript. As such, I support the publication of the present version of the manuscript.

We thank the reviewer for the kind words.

Reviewer #3

The authors have demonstrated commendable dedication in addressing both my feedback and that of other reviewers. Consequently, the manuscript has undergone significant enhancements, making the results and analysis considerably more robust. I am pleased with the overall improvements achieved in this new version.

The following review might be cited in the introduction while discussing computational methods for estimating quality and clinical relevance of cancer cell lines: PMID: 35822563

We thank the reviewer for the commendation. We have referred to this very recent paper on page 3.

References

- 1 Yu, K. *et al.* Comprehensive transcriptomic analysis of cell lines as models of primary tumors across 22 tumor types. *Nature Communications* **10**, 3574, doi:10.1038/s41467-019-11415-2 (2019).

Reviewers' Comments:

Reviewer #1:

Remarks to the Author:

The manuscript by now has been significantly improved and major issues have been resolved.

REVIEWERS' COMMENTS

Reviewer #1 (Remarks to the Author):

The manuscript by now has been significantly improved and major issues have been resolved.

We thank the reviewer for the positive comments.